

**Modeling the aging process of black carbon during atmospheric**
**transport using a new approach: a case study in Beijing**
Yuxuan Zhang[1,2], Meng Li[2], Yafang Cheng[2], Guannan Geng[3], Chaopeng Hong[4], Haiyan Li[1], Xin Li[1], Dan
Tong[4], Nana Wu[1], Xin Zhang[1], Bo Zheng[5], Yixuan Zheng[6], Yu Bo[1,7], Hang Su[2], and Qiang Zhang[1]
[1]Department of Earth System Science, Tsinghua University, Beijing 100084, China
[2]Multiphase Chemistry Department, Max Planck Institute for Chemistry, Mainz 55128, Germany
[3]Department of Environmental Health, Rollins School of Public Health, Emory University, Atlanta, Georgia 30322, USA
[4]Department of Earth System Science, University of California, Irvine, California 92697, USA
[5]Laboratoire des Sciences du Climate et de l'Environnement LSCE, Batiment 706, Pte 25, Orme de Merisiers 91191 Gif-sur-
Yvette, France
[6]Department of Global Ecology, Carnegie Institution for Science, CA, 94305, USA
[7]RCE-TEA, Institute of Atmospheric Physics, Chinese Academy of Science, Beijing, 100029, China
*Correspondence to*: Qiang Zhang (qiangzhang@tsinghua.edu.cn) and Yu Bo (boyu@mail.iap.ac.cn)
**Abstract.** The effect of black carbon (BC) on air quality and the climate is still unclear, which is partly because of the poor
understanding regarding the BC aging process in the atmosphere. In this work, we developed a new approach to simulate the
BC mixing state (i.e., other species coated on the BC surface) based on an emissions inventory and back-trajectory analysis.
The model tracks the evolution of the BC aging degree (characterized by the ratio of the whole particle size and BC core)
during atmospheric transport. Using the models, we quantified the mass-averaged aging degree of total BC particles transported
to a receptor (e.g., an observation site) from various emission origins (i.e., 0.25º×0.25º grids). The simulations showed good
agreement with the field measurements, which validated our model calculation. Modeling the aging process of BC during
atmospheric transport showed that it strongly dependent on emission levels. BC particles from extensive emission origins (i.e.,
polluted regions) were characterized by a higher aging degree during atmospheric transport due to more co-emitted coating
precursors. On the other hand, high-emission regions also controlled the aging process of BC particles that were emitted from
cleaner regions and passed through these polluted regions during atmospheric transport. The simulations identified the
important roles of extensive emission regions in the BC aging process during atmospheric transport, implying the enhanced
contributions of extensive emission regions to BC light absorption. This revealed that emission reductions in polluted regions
could achieve more benefits for improving air pollution and climate change. Emission reductions in polluted regions not only
decreased the aging degree of BC emitted from these regions but also reduced the aging process of BC emitted from other
origins during atmospheric transport. Moreover, emissions reduction in polluted regions may be more efficient in counteract
the suppression of planetary boundary layer (PBL) by BC particles (e.g., Zdunkowski et al. 1976; Jacobson 1998; Wendisch
et al. 2008; Barbaro et al. 2013; Ding et al. 2016; Wang et al., 2018), because a greater decrease in the BC aging degree during
atmospheric transport would weaken the light absorption capability of BC in the upper PBL. The simulation of the BC aging
degree during atmospheric transport provided more clues for improving air pollution and climate change.





## 1 Introduction

Black carbon (BC) plays an important role in the global warming and deterioration of air quality (Bond et al. 2013). The effects of BC aerosols on air quality and climate strongly depend on their light absorption. Accurately assessing the radiative effects of BC aerosols continues to be a major challenge in atmospheric/climate sciences, partly due to unclear light absorption capability of ambient BC particles. Estimates of the light absorption enhancement for BC-containing aerosols caused by coating materials on BC surface differ by a factor of ~3.5, spanning over a wide range from 1.05-3.5 (Cappa et al., 2012; Jacobson, 2001; Moffet et al., 2009; Peng et al., 2016). Generally, the aged BC during atmospheric transport exhibit a stronger absorption capability compared to near-source BC aerosols (Dahlkötter et al., 2014; Gustafsson and Ramanathan, 2016). In some climate model studies, the light absorption properties obtained from near-source BC aerosols are taken to estimate the direction radiative forcing (DRF) of BC (X. Wang et al., 2014; Schulz et al., 2006; Myhre et al., 2009). However, the climate effects of BC aerosols are on the regional to even global scales. Meanwhile, the effect of BC on air quality by the suppression on planetary boundary layer (PBL) is associated with atmospheric aging process of BC particles (Ding et al., 2016; Wang et al., 2018). Therefore, better understanding light absorption properties during atmospheric transport can improve the model prediction of BC effects on climate and air quality.

The change of light absorption of BC-containing particles during atmospheric transport is associated with the evolution of their mixing state. During atmospheric transport from emission sources, BC can internally mix with other atmospheric species (e.g., sulfate, nitrate, secondary organic matter and named coating materials) by condensation and coagulation processes. The interaction between BC and other aerosol components is defined as the BC aging process. Jacobson, (2001), Cheng et al., (2006), Lack and Cappa, (2010), Liu, et al., (2015) and Zhang et al., (2016) pointed out that aged BC can exhibit light absorption amplification by 2-3 times due to the lensing effect of the coating material on the BC surface, which then also influences the DRF and impact on the development of PBL (Cheng et al., 2008, 2009; Chung et al., 2012; Moffet et al., 2009; Ramanathan and Carmichael, 2008; Wendisch et al., 2008; Nordmann et al., 2014; Ding et al., 2016). However, the light absorption capability of BC during the aging process is still under debate (Cappa et al., 2012; Jacobson, 2001; Liu et al., 2017) partly due to a lack of understanding of the BC mixing state during atmospheric transport.

The mixing state of atmospheric BC-containing particles can be quantified by field observations, aircraft measurements and model simulations. Field measurements obtain the mixing state of BC particles as they are transported to an observation site (Cheng et al., 2006 and 2012; Moffet et al., 2009; Sedlacek et al., 2012; Zhang, et al., 2018a and 2018c). These observations characterize the average mixing state of BC over the observation site and cannot distinguish the mixing state of BC particles from different source origins. Moreover, field observations cannot be used to understand the temporal and spatial variations in the BC mixing state during atmospheric transport. Aircraft measurements are applied to explore the evolution of the BC mixing state during atmospheric transport from emission sources (Dahlkötter et al., 2014; Ditas et al., 2018; McMeeking, et al., 2011; Moteki et al., 2007). However, aircraft measurements are currently limited due to high costs, especially in developing countries with high BC emissions (e.g., China and India). On the other hand, some models have been developed to simulate the mixing state of BC-containing aerosols based on the mass or volume concentrations of BC and non-BC components (Jacobson, 2001; Matsui, et al., 2013; Oshima, et al., 2009). The major challenge of these models is how to treat non-BC components as coating materials of BC and BC-free particles. Moreover, when using these models, the computational cost is high when simulating the BC mixing state during atmospheric transport (Matsui, et al., 2013). A lack of information on the BC mixing state during the transport process will prevent a good understanding of the light absorption of BC particles in the atmosphere.

Considering the important contribution of BC from polluted regions to the BC amount present in the regional atmosphere (Lu et al., 2012; Zhang, et al., 2018c), studies on BC aging during atmospheric transport should pay more attention to BC particles from polluted regions. Based on in situ measurements, Zhang et al., (2018c) found that the light absorption capability of BC increased with increasing levels of air pollution due to more coating materials of BC under more polluted conditions.



Cheng et al. (2012) showed that the aging process in polluted areas, such as Beijing, was much faster than that in clean or less
polluted regions (Moteki et al., 2007; Shiraiwa et al., 2007) and in modeling studies (Cooke and Wilson, 1996; Cooke et al.,
1999, 2002; Lohmann et al., 2000; Jacobson 2001; Koch, 2001). Moreover, Peng et al., (2016) also pointed out a higher aging
rate of BC particles under more polluted environments. These studies identified the importance of understanding the mixing
state of BC during atmospheric transport from polluted regions.

6        In this work, we developed a new approach to simulate the evolution of the BC mixing state during atmospheric transport

based on BC emission inventory and back-trajectory analyses. First, the model calculation was used to simulate the mass-
weighted mixing state of overall BC particles at a Beijing site with a fine temporal resolution of one hour, which was compared
with the in situ measurements to evaluate our models. We then used our model to separate the mixing state of BC-containing
particles from various spatial origins (0.25º×0.25º resolution) as they were transported to a receptor site in Beijing during a
pollution period in late autumn. Based on the simulations, we focused on the mixing state of BC from polluted regions and
discussed the dependence of the BC mixing state during atmospheric transport on emission levels. Finally, we explored the
implication on the BC light absorption during atmospheric transport, especially for BC from polluted regions.
**2 Data and methods**
**2.1 Data**
**2.1.1 Measured aging degree of BC**
In this study, the aging degree of BC particles was characterized by the $D_p/D_c$ ratio (i.e., the ratio of the whole particle (including
coatings and BC core) and the BC core). The observed $D_p/D_c$ ratio of BC-containing particles is measured using a single-
particle soot photometer (SP2) in this work. The SP2 technique (Droplet Measurement Technology, Boulder, CO, USA) has
been described in detail elsewhere (Sedlacek et al., 2012; Moteki and Kondo, 2010; Zhang et al., 2016 and 2018b). In brief,
SP2 uses incandescence and scattering signals induced by a Nd:YAG intracavity laser beam at 1064 nm to quantify the
refractory BC (rBC) mass and the scattering cross section of individual BC-containing particles. The rBC mass determined
from incandescence signal of SP2 was calibrated with Aquadag particles of known masses. Details on SP2 calibration was
shown in previous studies (Zhang et al., 2018c). To retrieve the scattering cross section of BC-containing particles from
scattering signals of SP2, a leading edge only (LEO) fit method was used (Gao et al., 2007). $D_p$ and $D_c$ were derived from the
SP2 measurement and Mie theory, as given by Zhang et al. (2016, 2018c). In Mie calculation, the refractive indices of rBC
core and coating materials was prescribed the values of 2.26-1.26i (Taylor et al., 2015; Zhang et al., 2018b) and 1.50-0i (Cappa
et al., 2012; Zhang et al., 2018c); the density of rBC core was used as the value of 1.8 g cm$^{-3}$ (Cappa et al., 2012; Taylor et al.,
2015). The hourly mass-average $D_p/D_c$ ratio of BC-containing particles is used in this work.

30       Table 1 lists the field observations: BJNOV2014 (performed on 17-30 November 2014), BJOCT2014 (performed from 28

October to 2 November 2014), BJSEP2015 (performed on 12-19 September 2015) and BJAUG2015 (performed on 17-23
August 2015). The observation site (40º00'17" N, 116º19'34" E, shown in Fig. 2a) is located at Tsinghua University in the
downtown area of Beijing and can be representative of the urban environment (Zhang et al., 2018a and 2018c). In this study,
the Tsinghua site was taken as the receptor of BC particles from emission origins (e.g., Hebei, Tianjin, Shandong, Shanxi,
Shaanxi and Inner Mongolia, Fig. 2a) during atmospheric transport. The BJNOV2014 measurement contained several pollution
episodes (Fig. 1a), which featured the evolution of BC aging degree (i.e., $D_p/D_c$ ratio) associated with air pollution in the range
of 1.4-2.3 (Fig. 1b). During the BJNOV2014 campaign period, BC amount transported to the Tsinghua site was dominated by
the emission of Beijing and its surrounding areas (i.e., Hebei, Tianjin, Shanxi, Shaanxi and Inner Mongolia) (Fig. 3). Moreover,
Figure 3 shows that the spatial origins of BC over the observation site during the four the pollution episodes for the
BJNOV2014 measurement were different (e.g., urban, rural and industrial sources) based on back-trajectory analysis (Lu et





al., 2012). Thereby, the BC aging degree (i.e., $D_p/D_c$ ratio) obtained from BJNOV2014 measurement was representative and
were used to establish the model in this study. The other measurement periods (i.e., the BJOCT2014, BJSEP2015 and
BJAUG2015 cases) were characterized by the evolution of pollution episodes (i.e., from clean hours to slight pollution and
then reaching to pollution period and finally retuning to clean hours), which were used to identify that the models could
simulate BC aging degree with a high time resolution (i.e., an hour) to characterize the change of BC mixing state associate
with air pollution.
**2.1.2 Back-trajectory analysis and BC emissions**
During atmospheric transport, information on the location, height and transport time of BC-containing particles is obtained
from back-trajectory analysis using the Hybrid Single-Particle Lagrangian Integrated Trajectory (HYSPLIT) model, with the
meteorological fields from the National Centers for Environmental Prediction (NCEP) Global Data Assimilation System
(GDAS). Back-trajectories with an hourly temporal resolution were calculated. The arrival height was set as 100 m. We ran
the trajectories backwards for 3, 5 and 7 days. The effective amount of BC transported to the observation site derived from the
5 day back-trajectories was similar to that from the 7 day back-trajectories, which was significantly larger than that from the
3 day back-trajectories (Fig. 1c). This result indicated that the atmospheric lifetime of BC during the campaign period was ~5
days. Therefore, the 5 day back-trajectories were used in the following model calculation.
The gridded BC emissions for the years 2014 and 2015 were obtained from the MIX inventory, with a resolution of
0.25º×0.25º (http://www.meicmodel.org/dataset-mix). The MIX inventory includes the emission data of anthropogenic sources
in Asia (Li et al., 2017). Fig. 2b shows the gridded BC emissions at 0.25º×0.25º in Beijing and its surrounding regions (i.e.,
Hebei, Tianjin, Shandong, Shanxi, Shaanxi and Inner Mongolia).
**2.2 Model development of BC aging during atmospheric transport**
The BC aging process during atmospheric transport depends on the formation of coating materials (i.e., other species (e.g.,
sulfate, nitrate and organics) on the BC surface by condensation and coagulation). The more coating materials there are on the
BC surface, the more aged BC during atmospheric transport. The quantity of coating materials during atmospheric transport
strongly depends on the pollutant emission levels and BC transport time. In this study, the rate of change in coating mass
($m_{coating}$) on BC is defined as:
$\frac{dm_{coating}}{dt} = k_{aging}E$        (1)
where $t$ represents the transport time; $k_{aging}$ represents the aging rate coefficient; and $E$ represents the emissions level of coating
precursors, which are co-emitted with BC. To simplify the calculation, $E$ is quantified by BC emissions from the MIX
inventory. The variable $m_{coating}$ is calculated by Eq. (2):
$m_{coating} = \frac{1}{6}\pi D_p^3 \rho_p - \frac{1}{6}\pi D_c^3 \rho_c$        (2)
where $\rho_p$ and $\rho_c$ represent the densities of the total BC-containing particles (including coating materials and BC core) and the
BC cores only, respectively.
Combining Eqs. (1) and (2), the BC mixing state (i.e., the $D_p/D_c$ ratio) during atmospheric transport can be calculated as:
$(\frac{D_p}{D_c})^3 = kE_{aver}t + (\frac{D_p}{D_c})^3_{t=0}$        (3)
where $E_{aver}$ represents the average BC emissions during transport and $k$ represents the normalized aging rate coefficient, which
is expressed as:
$k = \frac{6k_{aging}}{\rho_p \pi D_c^3}$        (4)
Following Eq. (3), we calculated the $D_p/D_c$ ratio of BC aerosols transported to the receptor site from different source origins
with a 0.25º×0.25º resolution (i.e., the origin-resolved mixing state of BC). The conceptual scheme of the evolution of the BC



mixing state (i.e., the $D_p/D_c$ ratio) during atmospheric transport is shown in Fig. 4. When BC aerosols emitted in a grid, $h$, were
transported to the receptor site following a trajectory, $l$ (Fig. 4), the $D_p/D_c$ ratio (i.e., $(\frac{D_p}{D_c})_{h,l}$) is given, as shown in Eq. (5):
$$(\frac{D_p}{D_c})^3_{h,l} = kE_{aver,h,l}t_{h,l} + (\frac{D_p}{D_c})^3_{ini} \qquad (5)$$
where $E_{aver,h,l}$ and $t_{h,l}$ represent the average BC emissions (unit of t/grid/year) and BC transport time (unit of h) from the grid $h$
to the receptor site following the trajectory $l$, respectively, and $(\frac{D_p}{D_c})_{ini}$ represents the initial value of the $D_p/D_c$ ratio of BC
before transport (Fig. 4), which characterizes BC aging from the emission source to the top of the planetary boundary layer
(PBL) of the source origin (i.e., grid $h$).
For a trajectory, $l$, BC particles pass through a series of grids (i.e., $h_1$, $h_2$, $h_3$,…) to the receptor (i.e., $hn$), the average $D_p/D_c$
ratio (i.e., $(\frac{D_p}{D_c})_l$) of overall BC particles transported to the receptor site from various source origins (i.e., $h_1$, $h_2$, $h_3$,…$h_i$,…$h_n$)
is determined by the mass-weighted $(\frac{D_p}{D_c})_{h,l}$, which is expressed by Eq. (6):
$$(\frac{D_p}{D_c})_l = \sum_{h=1}^{TNG_l} [(\frac{D_p}{D_c})_{h,l} \times W_{h,l}] \qquad (6)$$
where $TNG_l$ represents the total number of contributing grids to BC over the receptor site following the trajectory $l$; $W_{h,l}$
represents the weighting factor of BC from the grid $h$, which is determined by the effective emission intensity (EEI; related to
the BC emissions and transport ability (i.e., hydrophilic-to-hydrophobic conversion and removal process) of BC from the
source origin to the receptor site) and was developed by Lu et al. (2012). $W_{h,l}$ is calculated by Eq. (7):
$$W_{h,l} = \frac{EEI_{h,l}}{\sum_{h=1}^{TNG_l} EEI_{h,l}} \qquad (7).$$
Following the algorithm developed by Lu et al. (2012), the EEI of BC transported the receptor site from the surface grid $h$
following a trajectory $l$ (i.e., $EEI_{h,l}$ in Eq. (7)) can be determined by Eq. (8):
$$EEI_{h,l} = E_h \times TE_{h,l}, \qquad (8)$$
where $TE_{h,l}$ represents the BC transport efficiency following the trajectory $l$, which is calculated following Eqs. (1)-(4) shown
in Lu et al. (2012).
Combining Eqs. (5), (6) and (7), the $D_p/D_c$ ratio of BC aerosols transported to the receptor site following trajectory $l$
(i.e., $(\frac{D_p}{D_c})_l$) can be calculated by Eq. (9), which quantifies the aging degree of the overall BC transported to the receptor site,
with a temporal resolution of 1 hour.
$$(\frac{D_p}{D_c})_l = \sum_{h=1}^{TNG_l} \left[ \left( kE_{aver,h,l}t_{h,l} + (\frac{D_p}{D_c})^3_{ini} \right)^{\frac{1}{3}} \times \frac{EEI_{h,l}}{\sum_{h=1}^{TNG_l} EEI_{h,l}} \right] \qquad (9)$$
The model parameters are the initial value of the $D_p/D_c$ ratio of BC at the top of the PBL (i.e., $(\frac{D_p}{D_c})_{ini}$ in the model calculations)
and the BC aging rate coefficient (i.e., $k$ in the models) during the atmospheric process need to be retrieved to determine BC
aging degree during atmospheric transport.
The $(\frac{D_p}{D_c})_{ini}$ value is estimated by the BC aging near different emission sources. In this work, the $D_p/D_c$ values of BC near
the industrial, residential and traffic sources were prescribed as 1.4, 1.6 and 1.2 (Liu et al., 2014; Liu et al., 2017; Liu et al.,
2015; Laborde et al., 2013; Healy et al., 2015; Kondo et al., 2011; Morgan et al., 2019; Pan et al., 2017; Ramnarine et al.,
2018; Willis et al., 2016; Schwarz et al., 2008; Shi et al., 2019; Wang et al. 2018). The $(\frac{D_p}{D_c})_{ini}$ at the source origin (i.e., grid h)
was taken as the mass-weighted values of different source types. Figure S1 shows that the $(\frac{D_p}{D_c})_{ini}$ values in Beijing and its



surrounding areas were dominated by 1.45-1.55, which was agreed with the lowest 5th percentile of the observed $D_p/D_c$ ratio
of ambient BC-containing particles at a Beijing site (Table 1). This validated the $(\frac{D_p}{D_c})_{ini}$ value used in the model calculation.

3        The aging rate coefficient $k$ is retrieved from in situ measurements. The $k$ value can be determined with an assumption that

the simulated $D_p/D_c$ ratio of BC-containing particles was equal to the measured one. It was noted that the simulated $(\frac{D_p}{D_c})_l$
values, with an hourly temporal resolution, from Eq. (9) might not be equal to the observed values at a certain hour because
some BC particles transported were not transported out of the observed site within one hour. To reduce the influence of the
incomplete dispersion of ambient aerosols, the experimentally determined $k$ was calculated based on the observed $D_p/D_c$ ratio
of BC-containing particles during a period $((\frac{D_p}{D_c})_{obs,p})$, such as the mass-average values during a pollution episode and the
whole campaign. In this work, the aging rate coefficient $k$ was calculated with the assumption that the $(\frac{D_p}{D_c})_{obs,p}$ values was
equal to the simulated $D_p/D_c$ ratio of BC-containing particles over the receptors from various source origins during a period
following various trajectories (i.e., $(\frac{D_p}{D_c})_p$).

12       Following Eqs. (5)-(7), $(\frac{D_p}{D_c})_p$ is calculated using Eqs. (10)-(12):

$$\left(\frac{D_p}{D_c}\right)_p = \sum_{h=1}^{TNG_p} [(\frac{D_p}{D_c})_{h,p} \times W_{h,p}] \tag{10}$$

$$\left(\frac{D_p}{D_c}\right)_{h,p} = \left( k \left( \frac{\sum_{l=1}^{TNT_{h,p}} E_{aver,h,l} t_{h,l}}{TNT_{h,p}} \right) + \left(\frac{D_p}{D_c}\right)_{ini}^3 \right)^{\frac{1}{3}} \tag{11}$$

$$W_{h,p} = \frac{EEI_{h,p}}{\sum_{h=1}^{TNG_p} EEI_{h,p}} \tag{12}$$

where $(\frac{D_p}{D_c})_{h,p}$ represents the $D_p/D_c$ ratio of BC-containing aerosols transported to our observation site from a grid, $h$, during
the period; $W_{h,p}$ represents the weighting factor of BC from the grid $h$ to the receptor site during the period; $TNG_p$ represents
the total number of contributing grids to BC over the receptor site during the period; $TNT_{h,p}$ represents the total number of
trajectories passing through the grid $h$ during the period; and $EEI_{h,p}$ represents the $EEI$ in the surface grid $h$ during our a period,
which is calculated by Eq. (13) (Lu et al., 2012):
$EEI_{h,p} = E_h \times [(\sum_{l=1}^{TNT_{h,p}} TE_{h,l})/TNT_p]$ (13)

20       where $TNT_{h,p}$ represents the total number of trajectories passing through the grid $h$ and $TNT_p$ represents the total number

of back-trajectories originating at our site during the whole period. Fig. 3 shows the TE density (TED) (defined as
$(\sum_{l=1}^{TNT_{h,p}} TE_{h,l})/TNT_p$ in Eq. (13), Lu et al., 2012) and EEI on a 0.25°×.25° grid during the episodes 1-4 (Fig. 1a). The gridded
EEI analysis revealed that BC particles over the Tsinghua site during the four episodes are from different source types, namely
industrial and rural sources, urban and rural sources, urban and industrial sources as well as industrial source during episodes
1-4, respectively. This indicated that the model parameters obtained from the BJNOV2014 measurement are representative.

26       With the assumption that $(\frac{D_p}{D_c})_p = (\frac{D_p}{D_c})_{obs,p}$, the experimentally determined $k$ was calculated by Eq. (14):

$$k = \left[ \left(\frac{D_p}{D_c}\right)_{obs,p}^3 - \left(\frac{D_p}{D_c}\right)_{ini}^3 \right] \times \frac{\sum_{h=1}^{TNG_p} EEI_{h,p}}{\sum_{h=1}^{TNG_p} \left[ \left( \frac{\sum_{l=1}^{TNT_{h,p}} E_{aver,h,l} t_{h,l}}{TNT_{h,p}} \right) \times EEI_{h,p} \right]} \tag{14}.$$

28       In this work, the $k$ value was calculated based on the measured $D_p/D_c$ ratios of BC-containing particles during BJNOV2014

campaign period at the Tsinghua site (40°00'17" N, 116°19'34" E) in Beijing. In order to obtain the $k$ value and then valid our
models, the BJNOV2014 measurements were divided into two parts: the first two episodes (i.e., 17-24 November 2014) used
to calculate the $k$ value and the last two episodes (i.e., 24-30 November 2014) used for model validation. Following the Eq.



(14), the normalized aging rate coefficient $k$ was determined to have a value of $1.8\times10^{-4}$ t$^{-1}$ h$^{-1}$, which was used to calculate the
hourly average $D_p/D_c$ ratio (i.e., $(\frac{D_p}{D_c})_l$ in Eq. (9)) and the gridded $D_p/D_c$ ratio of BC-containing particles over the observed site
during the whole campaign period (i.e., $(\frac{D_p}{D_c})_{h,p}$ in Eq. (11)).
To evaluate our model calculation of the mixing state of BC-containing particles over the site, the simulated $(\frac{D_p}{D_c})_l$ values
with an hourly temporal resolution are compared with the observed hourly $D_p/D_c$ ratio during 24-30 November 2014 (i.e., the
last two episodes during the BJNOV2014 measurement). It is noted that the observed $D_p/D_c$ ratio at a certain hour is not only
dominated by the aging degree of BC transported to the site at this time but also impacted by the aging degree of BC over the
site several hours beforehand due to the incomplete dispersion of ambient aerosols within one hour. In this work, we averaged
the simulated $(\frac{D_p}{D_c})_l$ at a certain hour and within n (n=0, 2, 4, 6….) hours beforehand ($(\frac{D_p}{D_c})_{aver,t}$) to compare with the observed
hourly $D_p/D_c$ ratio. $(\frac{D_p}{D_c})_{aver,t}$ is calculated by Eq. (15):

$$(\frac{D_p}{D_c})_{aver,t} = \frac{\sum_{l=t-n}^{t}(\frac{D_p}{D_c})_l}{n+1} \qquad (15).$$

Figure 5 shows an excellent agreement between the simulated and observed hourly $D_p/D_c$ ratio wth a difference of ~3%.
The simulated $(\frac{D_p}{D_c})_{aver,t}$ values, with n=2,4, and 6, exhibit a better correlation with the observed hourly $D_p/D_c$ ratio than those
without considering the effect of BC transported over our observation site within a few hours beforehand (i.e., n=0). When
$n=4$, the linear relationship between simulated and observed $D_p/D_c$ ratio exhibited a highest correlation coefficient (i.e.,
$R^2$=0.86), indicating that the residence time of BC-containing particles over the Tsinghua site during the campaign period (i.e.,
polluted periods) is ~5 hours.
**3 Results and discussion**
**3.1 Simulating BC mixing state with a high time resolution**
Following Eqs. (9) and (15), we calculated the hourly $D_p/D_c$ ratios of BC-containing particles during the BJOCT2014,
BJSEP2015 and BJAUG2015 measurement periods. Fig. 6 shows that the simulated $D_p/D_c$ values of BC-containing particles
exhibit significant changes with pollution levels (i.e., the PM$_{2.5}$ and rBC concentrations), revealing that our model with a high
time resolution can simulate the evolution of the BC aging degree during a pollution episode. The simulation results showed
that under pollution conditions, not only the BC mass concentrations were increased, but also the aging degree of BC-
containing particles enhanced. This identified that the amplification of BC light absorption associated with air pollution due to
more coating material on BC surface (Zhang et al., 2018c).
It is import for simulation of BC mixing state with a high time resolution to evaluate effect on air quality. The effect of BC
on air quality depends on both the mass concentration and aging degree of BC. Simultaneous increase in the mass concentration
and aging degree of BC associated with air pollution could suppress PBL by the dome effect (Ding et al., 2016). In China, the
air pollution often start rapidly within several hours (Zheng et al., 2016). It is necessary to understand hourly aging degree of
BC-containing particles for better exploring the effect of BC on air quality. Our model calculation can provide BC aging degree
(i.e., $D_p/D_c$ ratio) with a fine temporal resolution of an hour.
The simulated BC mixing state with high time resolution during different measurement periods was used to revealed that
our models could be generally applied. Although the model is established based on the BJNOV2014 measurement (17-30
November 2014, Beijing), the retrieved model parameters are applicable to other measurement periods (i.e., October 2014,
September 2015, and August 2015). Fig. 6 reveals an excellent agreement between the simulated and observed hourly $D_p/D_c$





ratio for the BJOCT2014, BJSEP2015 and BJAUG2015 measurement. During these measurement periods, the linear
relationship between the simulated $D_p/D_c$ ratio and the observed values shows slopes of 0.98-1.01, with a correlation coefficient
($R^2$) of 0.53-0.85. A good correlation further demonstrated the validity of our model for calculating the BC mixing state during
atmospheric transport. Accurate simulations for the different measurement periods identify the generalness of our model,
especially under polluted environments.
**3.2 BC mixing state during the atmospheric transport process**
**3.2.1 Aging degree of BC from emission origins to receptor sites**
Using our models, we investigated the aging process of BC-containing particles transported from different spatial origins to
the receptor (i.e., the Tsinghua site, Beijing) during the BJNOV2014 measurement period. The origin-resolved (0.25º×0.25º)
$D_p/D_c$ ratio of BC-containing particles over the receptor site calculated using Eq. (11), shown in Fig. 7. The BC-containing
particles from various spatial origins exhibited significant difference in their mixing state with $D_p/D_c$ ratio in the range of 1.4-
2.8 as they reached the Tsinghua site. The $D_p/D_c$ ratio of BC from the polluted regions (i.e., southern Heibei, northeastern
Hebei and Tianjin) could up to~2.3-2.8, which characterized the mixing state of fully aged BC in the North China Plain. These
fully aged BC particles played an important role in regional light absorption (Gustafsson and Ramanathan, 2016; Peng et al.,
2016). The high $D_p/D_c$ ratio (~2.3-2.8) of BC transported to our site was consisted with the mixing state of fully aged BC
particles ($D_p/D_c$ ratio ~2.5) in Beijing reported by Peng et al., (2016), implying reliable values of gridded $D_p/D_c$ ratio from our
model calculation.
In this work, we classified the spatial emission sources of BC-containing particles over the receptor site into local (or
Beijing (i.e., BJ origin), shown in Fig. 8a) and non-local (i.e., non-Beijing) origins based on political boundaries. The non-
local origins were further divided into Hebei, Tianjin and other origins (i.e., HB, TJ or OT origins, respectively, as shown in
Fig. 8a). Based on the gridded $D_p/D_c$ ratio shown in Fig.7, the $D_p/D_c$ ratios of BC-containing particles from Beijing, Hebei,
Tianjin and other regions during the campaign period were estimated to be 1.8, 2.2, 2.1 and 2.0, respectively (Fig. 8b). Fig.
2b reveals that more intensive emission regions of BC are located in southern Beijing, southern Hebei, northeastern Hebei and
Tianjin, which dominated BC amount over the receptor site during the campaign period (Fig. 3). The EEI analysis (Fig. 3)
shows that BC-containing particles transported to the site during the campaign period mainly originate from Beijing, Hebei,
Tianjin, Inner Mongolia, Shanxi and Shaanxi (the political boundaries of these regions are shown in Fig. 2a).
The contributions of different emission origins to the BC amount over the receptor site during the BJNOV2014
measurement period were estimated by EEI analysis. BC transported to the site was dominated by Hebei and Beijing as the
major source regions, accounting for ~40.2% and ~40.0% of the total amount transported, respectively. In terms of non-local
(i.e., non-Beijing) origins, the contribution of BC from Hebei was significantly higher than that from Tianjin (~4.2%) and
others (~15.5%) due to more intensive emissions and a larger region in Hebei Province. Approximately 60% of BC from non-
local origins (i.e., Hebei, Tianjin and others) indicated the importance of atmospheric transport to BC concentrations during
polluted environments. During the campaign period, the average BC mass concentration was ~4.0 μg m⁻³ (Fig. 1b). Based on
EEI analysis, the mass concentrations of local (i.e., Beijing) and non-local (i.e., non-Beijing) BC at the observed site were
estimated to be ~1.6 μg m⁻³ and ~2.4 μg m⁻³, respectively.
Atmospheric transport not only played an important role in BC mass concentration in Beijing under polluted environments
but also controlled BC aging. During the investigation period, the $D_p/D_c$ ratio of non-local (i.e., non-Beijing) BC over the site
was ~2.1, which was greater than that of local (i.e., Beijing) BC (~1.8). The higher aging degree of non-local BC could be
attributed to the longer transport time and larger emissions from non-local origins. As more intensive emission sources (i.e.,
polluted regions), Hebei and Tianjin were identified as the two largest contributing regions for the BC mixing state at the site
(Fig.7 and Fig. 8b). When BC particles were emitted from Hebei and Tianjin and then transported to the receptor site, their





$D_p/D_c$ ratios could reach up to 2.2 and 2.1, respectively, which were larger than that from other non-local origins (i.e., cleaner
regions). This could be due to more pollutant emission from southern Hebei, northeastern Hebei and Tianjin than that from
other regions (i.e., Inner Mongolia). The results revealed that BC particles emitted from polluted regions would exhibit a higher
aging degree during atmospheric transport, which is most likely attributed to more rapid aging due to more co-emitted coating
precursors (Peng et al., 2016).
On the other hand, high emission origins (e.g., southern Hebei, northeastern Hebei and Tianjin) also affect the aging process
of BC particles that pass through these regions. When BC particles are emitted from a clean origin and then pass through high
emission regions (e.g., southern Hebei) to the receptor site (cluster 3 shown in Fig. 8c), their $D_p/D_c$ ratio could reach ~2.2 (Fig.
8d). However, the emitted BC particles from the clean origin passing through a series of clean origins (cluster 1 shown in Fig.
8c) show a lower $D_p/D_c$ ratio (~1.6, Fig. 8d) as they reach the site. This difference identifies the important role of extensive
emission regions (e.g., southern Hebei) in the atmospheric aging process of BC particles emitted from other clean regions (e.g.,
Inner Mongolia). When BC particles passed through these polluted regions, their aging degree could be accelerated due to high
pollutant emission.
**3.2.2 Emission dependence**
As discussed above, the aging process of BC particles during atmospheric transport was closely associated with emissions
from regions through which they pass. To investigate the dependence of the BC mixing state during atmospheric transport, we
normalized the current emissions obtained from the MIX inventory as a unit (i.e., normalized emission of 1 shown in Fig. 9)
and set the scenarios of BC emissions reduced by 20%, 40%, 50%, 60%, and 80% (corresponding to normalized emissions of
0.8, 0.6, 0.5, 0.4, and 0.2, respectively, in Fig. 9).
Fig. 9a1 shows that the $D_p/D_c$ ratios of local (i.e., Beijing) BC, non-local (i.e., non-Beijing) BC and total BC (including
both local and non-local BC) transported to the site were proportional to the normalized emissions (i.e., the sum of both local
and non-local emissions), with slopes of 0.55, 0.26 and 0.44, respectively. The slope values revealed that the relative increase
or decrease in the $D_p/D_c$ ratio with emission change for non-local BC was ~2 times that of local BC. This result revealed that
the mixing state of non-local BC was more sensitive to emission change than that of local BC. Therefore, emission reduction
was more effective in lowering the aging degree of non-local BC compared with that of local BC. For example, when BC
emissions were reduced by 50% (i.e., from normalized emissions of 1 to 0.5 in Fig. 9a1), the $D_p/D_c$ ratios of non-local BC,
local BC and total BC at our site decreased by 14%, 7% and 11%, respectively. A greater reduction in the aging degree of non-
BC particles could be attributed to most of the non-local BC (~75%) from high emission origins (i.e., the southern Hebei,
northeastern Hebei and Tianjin regions). The origin-resolved $D_p/D_c$ ratio of BC transported to the site during the campaign
period shown in Fig. 7 indicated that a greater reduction in aging degree would be found for BC from higher emission regions,
such as southern Hebei, northeastern Hebei and Tianjin, which revealed the benefits of emission controls in extensive emission
regions. The results identified the dependence of the BC aging process during atmospheric transport on emissions, especially
for non-local BC from high emission origins.
To further evaluate the change in the mixing state of BC at the site from local (i.e., Beijing) and non-local (i.e., non-Beijing)
emissions, we simulated the origin-resolved $D_p/D_c$ ratio of BC over the receptor site only considering the changes in local (Fig.
9a2) or non-local (Fig. 9a3) emissions. It is noted that the mixing state of total BC at the receptor site not only depended on
the respective $D_p/D_c$ ratio of local and non-local BC but also their contributions to the total BC amount. When only altering
local or non-local emissions in the simulations, the contributions of various source origins to the BC amount at our site would
be changed.
When the changes in only local emissions were included in our simulations, the mixing state of total BC (including both
local (i.e., Beijing) and non-local BC (i.e., non-Beijing)) at the observed site was slightly sensitive to emissions. Fig. 9a2
presents the linear change in the $D_p/D_c$ ratio of total BC with changing local emissions (i.e., normalized emissions of 0.2-1),


with a slope of 0.07, which is significantly smaller than that (0.44) for the cases of changing both local and non-local emissions
(Fig. 8a1). Taking 50% of the reduction in local emissions as an example, the $D_p/D_c$ ratio of total BC at our site only decreased
by 2%. Meanwhile, 50% of the local emission reduction resulted in ~5.0% and ~4.6% of the decreases in the $D_p/D_c$ ratio for
local and non-local BC, respectively. The reductions in the $D_p/D_c$ ratio of local and non-local BC were larger than that in total
BC (~2%), which was due to the increase in the contributions of non-local BC (characterized by a larger $D_p/D_c$ ratio compared
to local BC) with a local emission reduction. The results showed that altering local emissions had a slight effect on the BC
mixing state at our site during the investigated period (i.e., under a polluted environment) due to slight change in the aging
degree of non-local BC.

9        In terms of non-local (i.e., non-Beijing) emission change, the response of the mixing state of total BC at our site to emission

changes was more significant than that for the case of local (i.e., Beijing) emission change. The linear correlation between the
$D_p/D_c$ ratio of total BC and normalized non-local emission (i.e., 0.2-1) shown in Fig. 9a3 exhibited a slope of 0.28, which was
markedly greater than that (0.07) in the case of local emission change (Fig. 9a2). Fig. 9a3 shows that the slope for total BC
(i.e., 0.28) was similar to that (i.e., 0.31) for non-local BC, indicating that the variation in the BC mixing state at our site during
the investigated period (i.e., under a polluted environment) was controlled by the emission change of non-local BC. This result
could be attributed to more aged BC particles were mainly from non-local regions (the southern Hebei, northeastern Hebei and
Tianjin). These results indicated that the BC mixing state at our site was dominated by non-local emissions (in particular
polluted regions with intensive emission), identifying the importance of atmospheric transport in the BC mixing state in Beijing
during polluted periods.
**3.3 Implication for BC light absorption**
BC light absorption depends on both the mass concentration and mixing state of BC. The light absorption of BC can be
characterized by multiplying EEI by the $D_p/D_c$ ratio (i.e., EEI*$D_p/D_c$). In this study, the origin-resolved EEI*$D_p/D_c$ values
represent the light absorption levels of BC particles as they were transported to the receptor site from various source origins
(0.25º×0.25º). Fig. 10a displays the origin-resolved EEI*$D_p/D_c$ values during the campaign period. High light-absorption levels
of BC were mainly from the local Beijing area, southern Hebei, and northeastern Hebei and Tianjin, resulting from high BC
emissions and strong BC aging in these regions.

26       The origin-resolved EEI*($D_p/D_c$) values revealed the contributions of BC from different source regions to light absorption

at the site (Fig. 10b). During the investigated period (i.e., polluted period) in Beijing, Hebei Province was the largest
contributing region, accounting for ~44% of BC light absorption transported to the observed site during the campaign period.
Local Beijing was responsible for ~36% of the light absorption of BC at the site, which was lower than the contribution from
Hebei. In total, ~64% of BC light absorption at the receptor site was contributed by non-local (i.e., non-Beijing) BC source
origins, reflecting the importance of atmospheric transport on the light absorption of BC in Beijing during polluted periods.
The contribution of non-local origins to BC light absorption at our site was larger than that to BC mass concentration (~60%,
as quantified by the EEI analysis), which was due to the higher aging degree ($D_p/D_c$ ratio of ~2.12) of non-local BC compared
to that of local BC ($D_p/D_c$ ratio of ~1.78). If the difference between the mixing states of local and non-local BC is not
considered, the effect of atmospheric transport on BC light absorption in Beijing during the polluted period would be
underestimated. The results revealed that the BC aging process during atmospheric transport strengthens the importance of
emissions from surrounding areas (e.g., Hebei and Tianjin) to BC light absorption in Beijing during polluted periods.

38       The strong dependence of both the mass concentrations and mixing states of BC on emissions indicates that emission

reduction could significantly lower light absorption and, thus, weaken the effect of BC on air quality and climate, especially
during polluted periods. A linear decrease in BC light absorption with emission reductions was found (Fig. 9b). When both
local (i.e., Beijing) and non-local (i.e., non-Beijing) BC emissions were reduced (Fig. 9b1), the change in light absorption for
non-local BC was much more significant than that for local BC. For the case of a 50% reduction in both local and non-local





BC emissions, the light absorption of total BC, local BC and non-local BC decreased by 55%, 53% and 56%, respectively
(Fig. 9b1). An extra decrease of ~6% in light absorption for non-local BC was greater than that (~3%) for local BC, which
could be attributed to a much greater reduction in the aging degree (i.e., $D_p/D_c$ ratio) of non-local BC (Fig. 9a1). Compared
with local BC, a much greater decrease in the light absorption of non-local BC implied that the emission reduction from non-
Beijing sources rather than Beijing sources achieved many more benefits in terms of the BC effect in Beijing. This result also
revealed that the emission reduction of surrounding areas of Beijing not only brought less BC but also lowered the BC mixing
state, which enhanced the decrease in BC light absorption in Beijing.

8        The analyses of only reducing local emissions or non-local emissions further identified the importance of emission

reduction of non-Beijing sources to weaken BC light absorption in Beijing during polluted periods. For the case of local (i.e.,
Beijing) emissions reduction (Fig. 9b2), the reduction in BC light absorption was significantly smaller than that for the case
of non-local (i.e., non-Beijing) emissions reduction (Fig. 9b3), revealing that emissions reduction in non-Beijing sources
played a more important role in decreasing BC light absorption in Beijing. For example, when emissions were reduced by 50%
for local and non-local origins, the BC light absorption at the site decreased by 21% and 35%, respectively. A greater decrease
in BC light absorption in Beijing with non-local emissions reduction resulted from larger contributions of non-Beijing source
emissions (especially for polluted regions, such as southern Hebei, northeastern Hebei and Tianjin) to both the mass
concentration and mixing state of BC over the site. Moreover, the extra reduction in BC light absorption in Beijing caused by
weakening of the BC aging degree was also greater under non-Beijing emissions reduction compared with that under Beijing
emissions reduction (e.g., ~5% and 1% extra reductions for the cases of 50% reductions in non-local and local emissions,
respectively). The import contribution of emission reduction of non-Beijing sources to decrease of BC light absorption in
Beijing under polluted environment was mainly due to significant decrease in both BC amount and its aging degree in Beijing
caused by emission reduction of polluted regions (e.g., southern Hebei, northeastern Hebei and Tianjin). This suggested that
the efforts to weaken the influence of BC on air pollution and climate change should pay more attention in emissions reduction
in polluted regions.
**4 Discussion**

25       In this study, we focused on investigating the aging process of BC during atmospheric transport under polluted conditions.

A rather simplified scheme was adopted where the aging rate is assumed proportional to the emissions without detailed
consideration of the effects of temperature, particle sizes, phase state, hygroscopicity and chemistry (Riemer et al. 2009; Cheng
et al. 2008, 2012, 2015; Mu et al. 2018). Actually, the aging rate coefficient ($k$) in our models was related to meteorological
factors (e.g., temperature, relative humidity (RH)), chemistry (e.g., oxidant concentrations), aerosol phase state (e.g.,
heterogeneous reaction) as well as other parameter (e.g., particle size). In this work, the experimentally determined $k$ value
was derived from the observed BC aging degree during the pollution periods. In terms of the pollution periods with low $O_3$
concentrations and high RH, the BC mass-weighted $O_3$ and RH levels at the observed site were 1-3 ppb and 50-63%,
respectively. The retrieved $k$ value used in the model calculations in this work was suitable to estimate BC aging under polluted
environment.

35       A higher aging degree of BC was found from more extensive emission regions during atmospheric transport. Our work

identified the important contribution of high emission regions to air pollution and climate change, highlighting the effect of
high emissions in these regions on the aging process of BC emitted from within that region and from other regions (i.e., low
emission regions). In terms of BC emitted from polluted regions (i.e., extensive emission origins), these regions were
characterized by higher aging rates during atmospheric transport due to more co-emitted coating precursors. On the other hand,
when BC particles from other origins passed through these high emission regions, their aging degree would be significantly
enhanced by speeding up the coating processes in polluted air, which revealed that for BC particles emitted from clean regions



(i.e., low emission areas), their mixing state during atmospheric transport under polluted environments was dominated by the
aging process when they passed through polluted regions (i.e., high emission areas). The influence of extensive emissions in
the polluted regions on the mixing state of BC from other regions enhanced the contributions of high emission areas to the BC
effect on air quality and climate.
Absorbing aerosol aloft increases atmospheric stability, which suppressed the development boundary layer and suppress
or enhance the formation of different types of clouds (Zdunkowski et al. 1976; Jacobson 1998; Wendisch et al. 2008; Barbaro
et al. 2013; Bond et al. 2013; Ding et al., 2016; Wang et al., 2018). More aged BC particles from high emission regions (e.g.,
southern Hebei) could enhance these effects as they are transported to other areas (e.g., Beijing) under polluted environments
because more aged BC exists in the upper PBL. Our simulations showed that the aging degree of BC from atmospheric
transport along with polluted air masses was significantly higher than that of local BC under polluted conditions, implying that
the light absorption capability of BC in the upper PBL would be higher than that of BC in the lower PBL. This characteristic
favored the formation of the inversion layer due to more heating in the upper PBL, which consequently depressed PBL
development. In China, air pollution generally occurs at the regional scale due to atmospheric transport (Sun et al., 2014; L.
Wang et al., 2014; Yang et al., 2015; Zheng et al., 2015). Regional pollution would bring more aged BC due to faster aging
processes during atmospheric transport. The enhanced effect caused by more aged BC from atmospheric transport would
further suppress the development of boundary layer (e.g., Zdunkowski et al. 1976; Jacobson 1998; Wendisch et al. 2008;
Barbaro et al. 2013; Ding et al. 2016) and together with the special haze chemistry (Cheng et al. 2016) strongly strengthens
regional pollution.
Our simulations reveal that emission reduction not only decreased the BC amount present in the atmosphere but also
weakened the BC aging degree during atmospheric transport. This pattern has been demonstrated by observations with and
without emission controls (Zhang et al., 2018a). The weakening of the BC aging degree during atmospheric transport was
dominated by emission reduction in polluted regions. Emissions reduction in polluted regions was not only responsible for
slowing the aging of BC emitted from these regions but also controlled the decrease in the coating materials of BC particles
passing through these polluted areas during atmospheric transport. Weakened aging processes decrease the light absorption of
BC during atmospheric transport. On the other hand, less aged BC from atmospheric transport could counteract the suppression
of PBL effect (e.g., Zdunkowski et al. 1976; Jacobson 1998; Wendisch et al. 2008; Barbaro et al. 2013; Ding et al. 2016) by
reducing the light absorption of BC in the upper PBL, in which BC particles mainly derive from atmospheric transport. Our
work identified that reducing emissions from polluted regions could achieve more benefits for air quality and the climate than
previously considered (i.e., lowering the BC mass concentration).
**5 Concluding remarks**
The effect of BC-containing particles on air quality and climate is not only dominated by BC mass concentration but also
controlled by their mixing state. To better understand the mixing state of atmospheric BC in China, we developed a new
approach to simulate the BC aging process during atmospheric transport. Our models track the BC mixing state (i.e., $D_p/D_c$
ratio) from an emitted source origin (e.g., a 0.25º×0.25º grid) to a receptor (e.g., the Tsinghua site in this study). The model
calculation can quantify the mass-averaged aging degree of overall BC particles over the receptor site from various origins.
The simulations can provide information on the BC mixing state with fine temporal and spatial resolutions. Based on the model
calculation, the regional climate effect of BC in China could be better understood. On the other hand, a high temporal resolution
up to one hour can display the evolution of the BC mixing state with pollution development, which plays an important role in
understanding the BC effect on air quality (Zhang et al., 2018c).
The models simulate the BC mixing state during atmospheric transport based on the BC emission inventory and back-trajectory
analysis. Two model parameters (i.e., the initial value of the $D_p/D_c$ ratio and the normalized aging rate coefficient $k$) were need





to be retrieved. The initial value of the $D_p/D_c$ ratio, which characterized BC aging from the emission source to the top of the
PBL over the source origin, was estimated by the BC aging near different emission sources. On the other hand, the normalized
aging rate coefficient $k$, with a value of ~$1.8\times10^{-4}$ t$^{-1}$ h$^{-1}$, was determined by the observed $D_p/D_c$ ratio of BC particles during
the BJNOV2014 measurement period. The established model was used to simulate the hourly $D_p/D_c$ ratio of BC-containing
particles, which agreed well with the observed ones. This good agreement validated our model calculation.
Using our model, we quantified the BC aging degree from various source origins to the receptor site (i.e., the Tsinghua site
in Beijing) during the BJNOV2014 measurement period (i.e., polluted period). The origin-resolved $D_p/D_c$ ratio of BC particles
over the site showed that the degree of aging (~1.78) of local (i.e., Beijing) BC was significantly less than that (~2.12) of non-
local (i.e., non-Beijing) BC during the campaign period. BC particles with higher aging degrees at our site mainly derived
from more intensive emission origins (e.g., southern Hebei) due to higher aging rates. On the other hand, when BC particles
emitted from clean origins passed through polluted regions (e.g., southern Hebei), they were also characterized by a greater
$D_p/D_c$ ratio by speeding up the aging process in polluted air. Our simulations revealed a strong dependence of BC mixing on
emissions during atmospheric transport. The mixing state of non-local (i.e., non-Beijing) BC particles was more sensitive to
emission change than that of local (i.e., Beijing) BC, especially for BC from non-local origins with high emissions. The mixing
state of total BC (including local and non-local BC) in Beijing was dominated by non-local emissions during the campaign
period. If only reducing local emissions, the mass-average $D_p/D_c$ ratio of total BC transported to Beijing would exhibit a slight
change (e.g., decreasing by ~2% with a 50% emission reduction). The response of the BC mixing state to emission change
identified the importance of atmospheric transport on the mixing state of BC particles in Beijing during the polluted period.
Our simulations of the BC mixing state provide an important implication for BC light absorption during atmospheric
transport. Atmospheric transport not only brought more BC but also enhanced the aging degree of BC over the site during the
polluted period. Considering the contributions of non-local (i.e., non-Beijing) emissions to both the mass concentrations and
mixing states of BC, atmospheric transport was responsible for ~64% of BC light absorption at our site during the campaign
period. Hebei was the largest contributing region for BC light absorption at the site, accounting for ~44%, followed by local
Beijing (~36%). The importance of atmospheric transport on BC light absorption in Beijing revealed that the effect of BC
aerosols on air quality and the climate should be evaluated at the regional scale. BC light absorption during atmospheric
transport could be affected by the emissions of areas that they passed through, especially the polluted regions (e.g., southern
Hebei). Emissions reduction in polluted regions not only significantly decreased the light absorption of BC emitted from these
regions but also weakens the light absorption of BC passing through these polluted areas due to lowering their rate of aging
during regional transport. Our simulations implied that mitigation efforts for both air pollution and climate change should well
understand the mixing state of BC particles during atmospheric transport, especially for those from polluted regions.
**Acknowledgments**
This work was supported by the National Natural Science Foundation of China (41571130032, 41571130035, 41625020, and

33    91744310).

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



1    Table 1 The in situ measurement in this work. $(D_p/D_c)_{5\%}$ represents the average value of the lowest 5th percentile of the hourly

2    $D_p/D_c$ ratio for ambient BC-containing particles during the campaign period.

| Measurement | Period | Location | Hourly PM$_{2.5}$ ($\mu g\ m^{-3}$) | Hourly rBC ($\mu g\ m^{-3}$) | Hourly $D_p/D_c$ | $(D_p/D_c)_{5\%}$ |
|---|---|---|---|---|---|---|
| BJNOV2014 | 17-30 November 2014 | Tsinghua site, Beijing | ~10-440 | ~0.2-14 | ~1.4-2.3 | 1.54 |
| BJOCT2014 | 28 October-2 November 2014 | Tsinghua site, Beijing | ~5-200 | ~0.1-7 | ~1.4-2.2 | 1.50 |
| BJSEP2015 | 12-19 September 2015 | Tsinghua site, Beijing | ~3-190 | ~0.1-4 | ~1.5-2.2 | 1.55 |
| BJAUG2015 | 16-23 August 2015 | Tsinghua site, Beijing | ~3-110 | ~0.1-3 | ~1.5-2.0 | 1.55 |





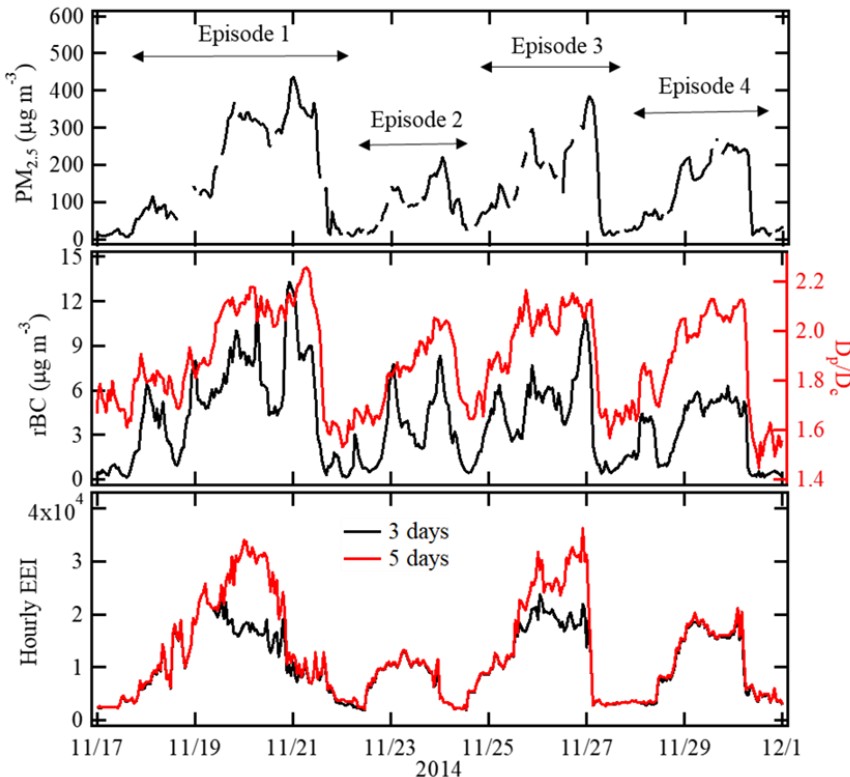

2    **Figure 1.** Time series of the PM$_{2.5}$ concentration, rBC mass concentration, $D_p/D_c$ ratio of BC-containing particles and the

3    hourly EEI for BC transported to the observation site for 3 and 5 days.





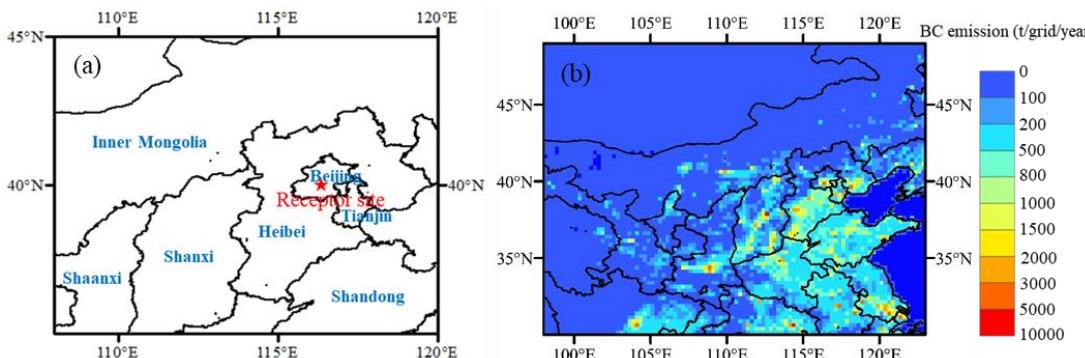

2 **Figure 2.** (a) The location of the receptor site (Tsinghua site (40º00'17" N, 116º19'34" E), marked as red star) during the

3 BJNOV2014 measurement period. (b) Spatial distribution (0.25°×0.25°) of BC emission inventory from MIX.



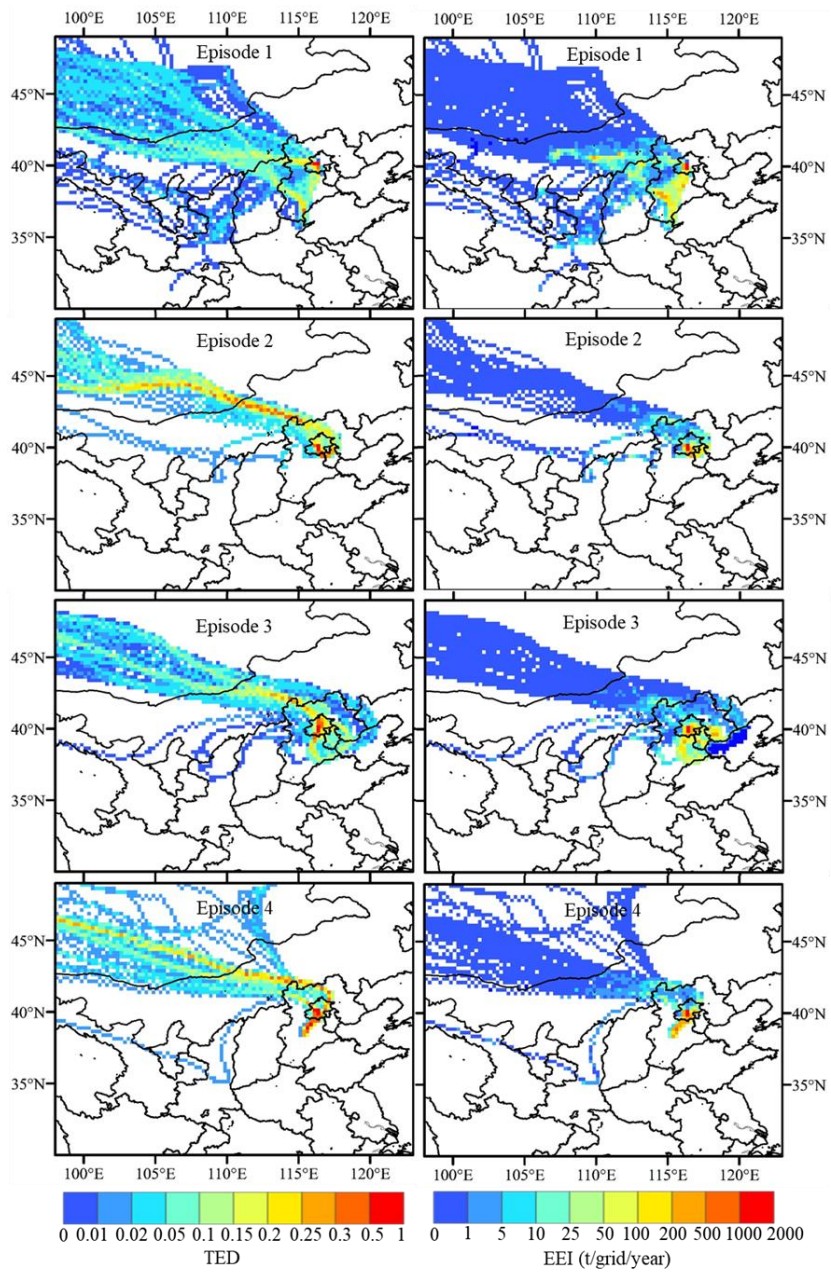

**Figure 3.** Spatial distribution (0.25°×0.25°) of the TED and EEI for BC transported to the observation site (40º00'17" N,
116º19'34" E) during episodes 1-4 of the BJNOV2014 measurement period. The TED and EEI were derived from BC emission
and black-trajectory analysis (Lu et al., 2012). The TED represents effective trajectory density transported to the receptor site.
The EEI quantifies the effective emission amount transported to the receptor site from emission origins.



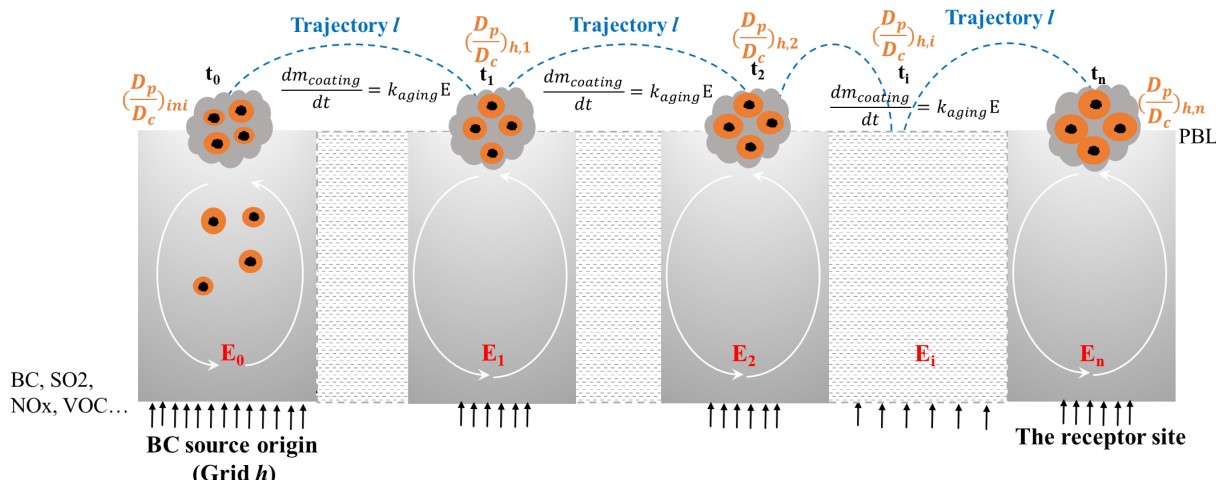

2    **Figure 4.** Conceptual scheme of the evolution of the mixing state (i.e., $D_p/D_c$ ratio) of BC from the source origin (i.e., grid $h$) to the receptor site following trajectory $l$.





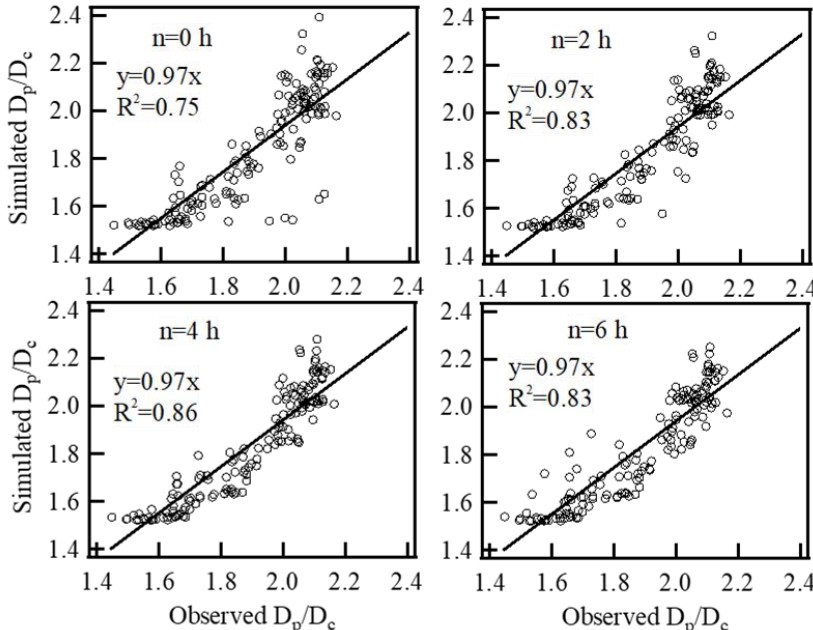

**Figure 5.** Comparison of the hourly $D_p/D_c$ ratio of BC between the model simulations and observations at our site with different

$n$ values in Eq. (15) on 24-30 November 2014 at Tsinghua site. The $n$ values represent the simulated hourly $D_p/D_c$ ratio

considering the effects of 0, 2, 4, and 6 hour backward transport using Eq. (15).



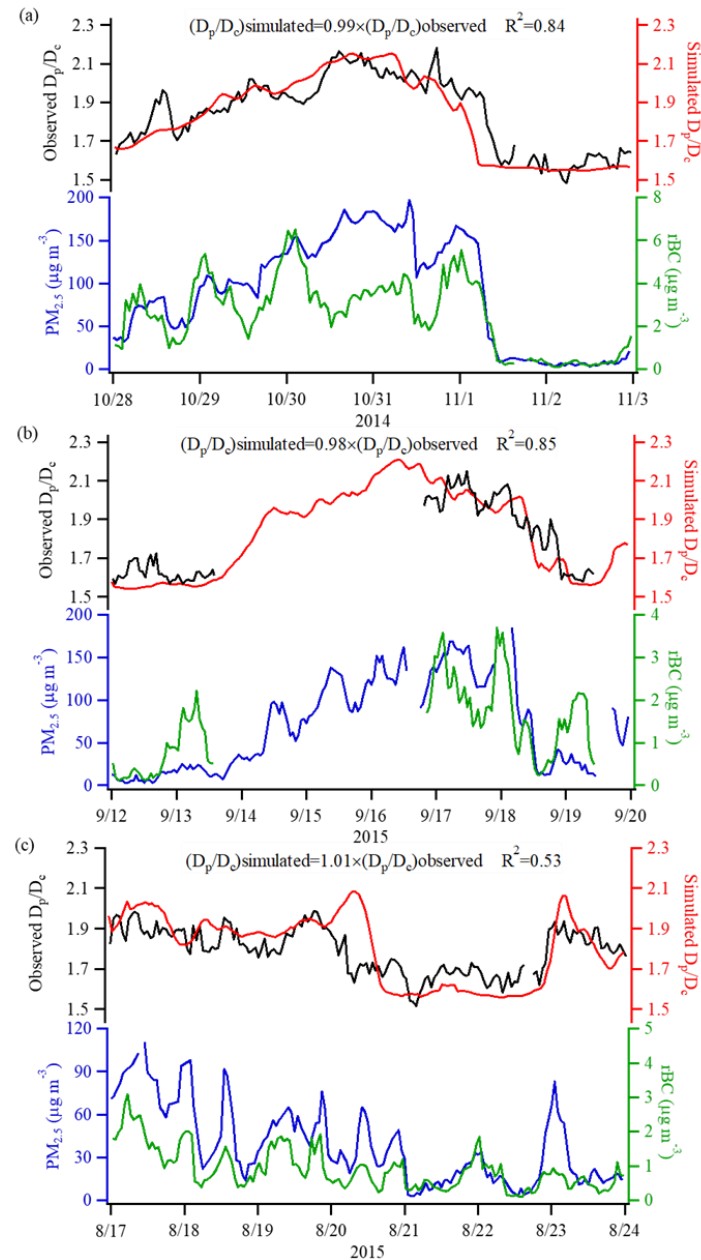

2    **Figure 6.** Time series of the observed $D_p/D_c$ ratio, simulated $D_p/D_c$ ratio, and PM$_{2.5}$ and rBC concentrations during the (a) BJOCT2014,

3    (b) BJSEP2015 and (c) BJAUG2015 measurement period.





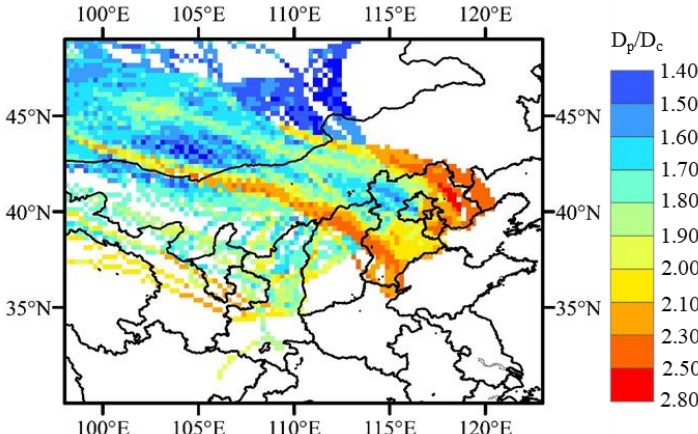

2    **Figure 7.** Aging degree (i.e., $D_p/D_c$) of BC particles as they are transported to the receptor site (40º00'17" N, 116º19'34" E)

3    from emission origins (0.25°×0.25°) during the BJNOV2014 campaign period. The gridded $D_p/D_c$ ratio was calculated by

4    Eq. (11).



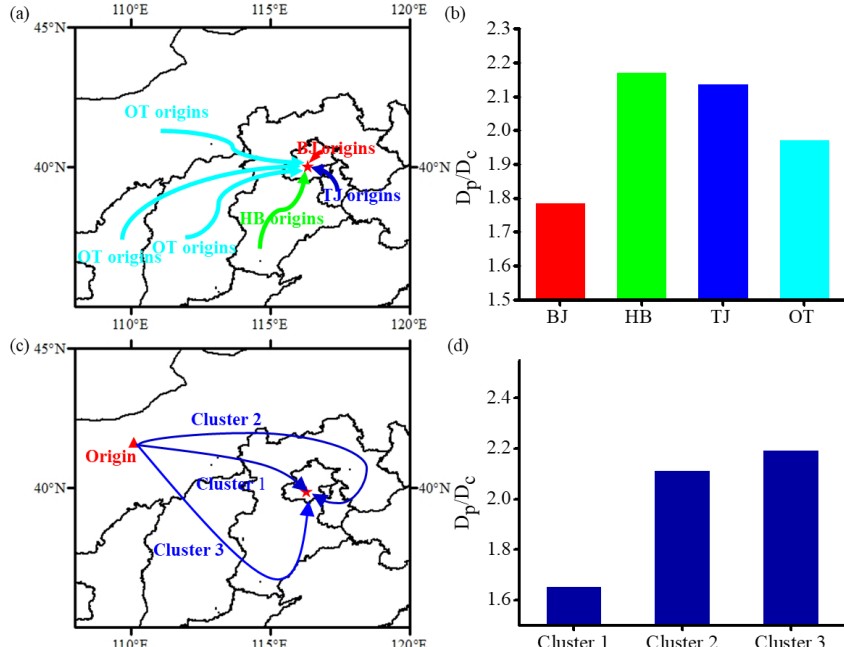

**Figure 8.** BC aging from emission origins to the receptor site (40º00'17" N, 116º19'34" E; marked with a red star) during the BJNOV2014
measurement period: (a) classification of the BC emission source regions based on political boundaries (i.e., the Beijing source regions (BJ
origins), Hebei source regions (HB origins), Tianjin source regions (TJ origins) and other source regions (OT origins)); (b) aging degree
(i.e., $D_p/D_c$) of BC particles emitted from the four source regions as they are transported to the receptor site; (c) conceptual scheme of
trajectories that pass through low emission regions (cluster 1), medium emission regions (cluster 2) and high emission regions (cluster 3) as
they transport from clean origins (i.e., Inner Mongolia) to the receptor site; and (d) the degree of aging of BC reaching the receptor site
through different trajectories (i.e., Clusters 1, 2 and 3 shown in (c)).




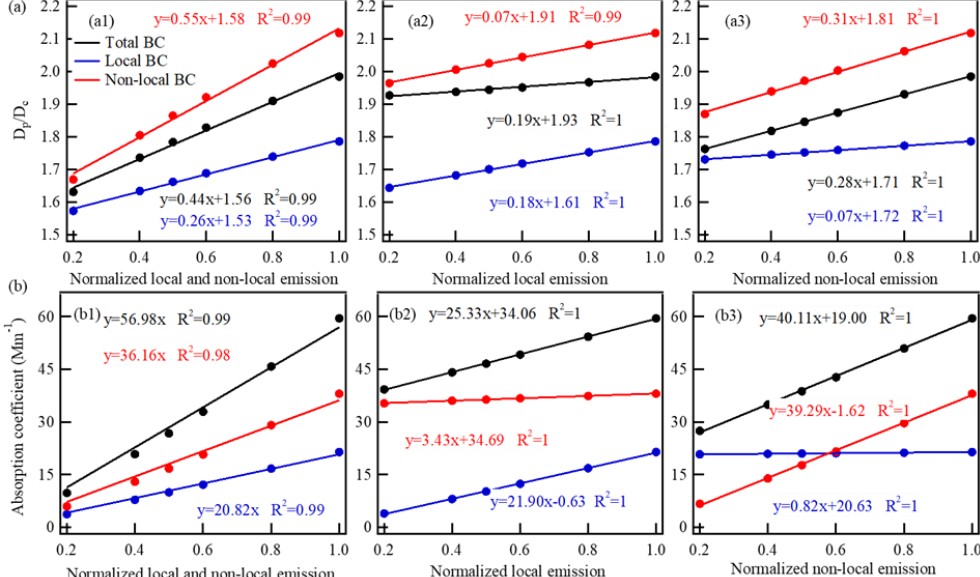

**Figure 9.** Variations in the (a) $D_p/D_c$ ratio and (b) light absorption coefficient for total, local and non-local BC over the site
with normalized emissions. The current emissions obtained from the MIX inventory (Fig. 2b) were normalized as a unit (i.e.,
normalized emission of 1), and the emissions reductions of 20%, 40%, 50%, 60% and 80% corresponded to the normalized
emissions of 0.8, 0.6, 0.5, 0.4, and 0.2, respectively. a1 and b1 represent the simulations for the case of both local and non-
local emission variations, respectively; a2 and b2 represent the simulations for the case of only local emissions variations; and
a3 and b3 represent the simulations for the case of only non-local emissions variations. The light absorption coefficient of BC
was estimated by the BC mass concentrations, the mass absorption cross section of BC (7.5 $m^2$ $g^{-1}$ at 550 nm) and the $D_p/D_c$
ratio. When the normalized emissions of total BC were equal to 1, the average mass concentration of total BC was ~4.0 µg m$^-$
$^3$, which was obtained by measurements during the campaign period. The mass concentration of local and non-local BC can
be further calculated based on their EEI contributions (i.e., ~1.6 µg m$^{-3}$ for local BC and ~2.4 µg m$^{-3}$ for non-local BC). Based
on the linear decrease in BC mass concentration with emission reduction, the mass concentrations of total, local and non-local
BC for different emission cases were calculated.





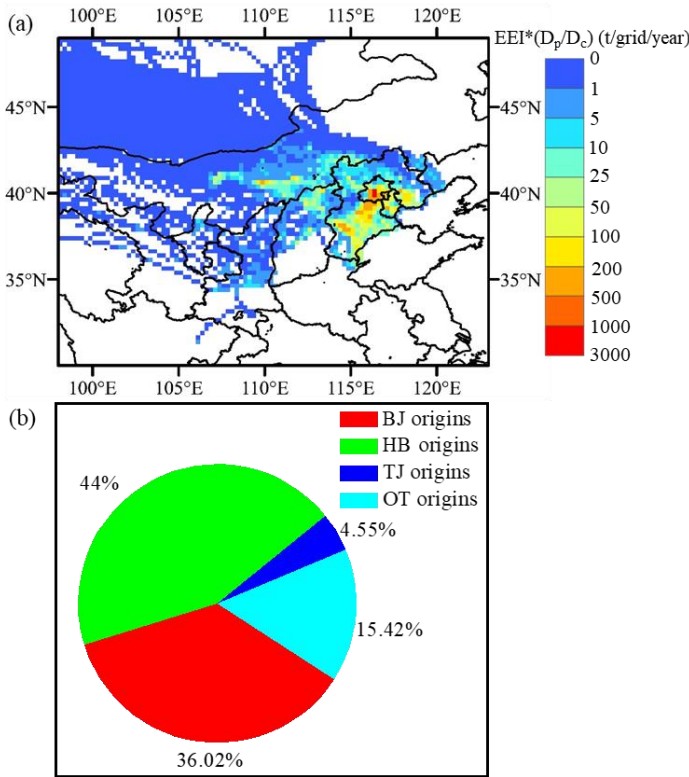

Figure 10. (a) Spatial distribution (0.25°×0.25° resolution) for light absorption levels (i.e., EEI*($D_p/D_c$)) of BC as it is
transported to the receptor site (40°00'17" N, 116°19'34" E) from various source origins. (b) Contributions of different source
regions to BC light absorption at the receptor site. The classification of source regions based on political boundaries is shown
in Fig. 8a. The BJ, HB, TJ and OT origins represent spatial sources of emitted BC transported to the receptor site from Beijing,
Hebei, Tianjin and other regions, respectively.