# Peer review of "Modeling the aging process of black carbon during atmospheric"

_Atmospheric Chemistry and Physics, 2019_

## Referee Comment (RC1) · Anonymous Referee #2 · 24 Apr 2019

Modeling 1 the aging process of black carbon during atmospheric transport using a new approach: a case study in Beijing

Zhang et al.,

The work developed a new approach to simulate the BC mixing state based on an emissions inventory and back-trajectory analysis. They quantified the mass-averaged aging degree of total BC particles and found the aging process of BC during atmospheric transport showed that it strongly dependent on emission levels. In general, from the modelling development to the modelling implication, I think that the result is worth to publish in the ACP after one minor revision.

[Figure]

Introduction P2L16-17 need references such as Li et al., JGR, 121(22), 13,784-713,798. P2L39 The author review several method to determine the BC particles. Also, Wang et al., ESTL, 2017 measure the Df in polluted air following soot particle aging. Does the model possibly consider the dimension fraction of BC? That would be more interesting. P7L24 deleted were P7L25 identified to indicated P7L27 import? Change the word P9L2-L3 Here it is possible to consider how RH influence BC coating? If the authors want to get the conclusion, firstly you need to exclude other possible factors..Therefore, as you try to do it. P9L6-13 Here some logical problem. if the aerosol in clean regions pass through the polluted area, how you know these BC don't deposit. How do you know BC at the sampling site were from clean region instead of the close polluted areas. Seemly, these are difficult questions about the mixture of air masses during the transport. How the authors separate them here? P10-P11, the implications and discussion should be combined? The implications should not be in Result section. The authors need to consider the structure in the two section. P12L10-11, capability should be individual BC or total here? My question is that the statement could be one problem. If you think total capability of BC in upper layer is higher that of BC in the lower. You need to know the BC distribution in the column within PBL. Normally the BC concentration is much higher than them in upper layer. Even mixing state is higher in upper layer but their concentration is lower. Therefore, the total capacity need be questioned. P12 section 5 concluding remarks. Most of them repeat the results or others. Is that possible to shorten it? The details should not be concluded here such as P13L2-5 and so on. The reference should be removed in the section.

---

## Referee Comment (RC2) · Anonymous Referee #1 · 26 Apr 2019

This paper shows a new and innovative method of trying to represent observed changes in the 'coating' of black carbon, as detected as changes in the Dp/Dc metric produced by the DMT SP2. This works by combining an emission field with a Langrangian transport model and an empirical parameterisation for the evolution of black carbon in the atmosphere. While this 'k' parameter is tuned to the measurements, the model shows an impressive correlation with the observations, giving confidence that there is value in the technique. The authors then go on to make estimates of the contributions from inside and outside the city to the measured concentrations and properties.

While not an all-encompassing process model, this does present an intriguing new way of looking at the data, that could conceivably bridge the gap between a highly detailed process model such as PARTMC-MOSAIC and Eularian chemical transport models like WRF-CHEM. It is also good that it provides another new perspective on the phenomenon of pollution 'building up' in Beijing, offering further evidence that this is driven by regional transport and transformation rather than local sources and processes.

While the main innovation with this paper is technical, the authors do use it to interpret air quality sources and phenomena, so I would say this is in-scope for ACP (as opposed to AMT or GMD). While I am not completely convinced of the results (see below), I think that the novelty of the technique alone means that it deserves to be discussed within the scientific literature. Overall, the quality of English is good, but I do have certain issues regarding how the results are interpreted and presented. As such, I recommend publication after revisions.

General comments:

While the earlier sections of the paper were well written, I was not impressed with how the paper interpreted the implications of the results. Sections 4 and 5 in particular seem to just largely restate what was already said in section 3 in different ways, so this entire portion of the paper could do with rewriting and sharpening up. But more generally, I would question what the key implications of this paper are; the conclusions seem to work off the reduction of Babs being a regulatory motivation, however most evidence supports BC being a more important metric for human health, which while related, is more directly related to emissions rather than processing. While the evolution of the mixing state of BC is important to consider for wet removal and radiative transfer, I would say it is debatable as to whether reducing the coatings (as opposed to overall BC) should be considered an objective for policymaking (while BC is recognised as a global climate forcing agent, the majority of this is likely from biomass burning). It is my opinion that the conclusions of this paper are more applicable to process-level atmospheric science, except for the part where the contributions to bulk BC are discussed.

I would recommend the discussion and conclusions be reframed accordingly.

Specific comments:

I found this paper extremely abstract and hard to read in places regarding certain quantities and it took me multiple reads before I think I began to understand what was going on. It particular, I would have appreciated a more intuitive explanation of what things like 'k' and 'EEI' physically represent (along with many others).

More discussion should be given to the possible mechanisms for the increase in Dp/Dc, i.e. coagulation and secondary aerosol formation. Given that there are already other models out there that consider these processes, is it possible to compare the value of the 'k' parameter to equivalent timescales in other models?

Ultimately, while good correlation between measurement and model is reached, it is possible that other yet-to-be-identified factors may be responsible for this agreement rather than the 'k' parameter being a good representation of ageing, which is the working hypothesis. The authors should spend more time discussing how best to further test the robustness of this model. In particular, a major limitation of this work is that it is restricted to autumn/winter datasets. During the summer in Beijing, there is typically much more photochemistry (reflected in high ozone concentrations) but BC concentrations and coating thicknesses are both reduced according to Liu et al. (https://www.atmos-chem-phys-discuss.net/acp-2018-1142/). Can this result be reconciled with this model? Would it mean that the 'k' value would need to have a seasonal dependence if this model were to be extended to other months?

---

## Author Comment (AC1) · 24 Jun 2019

We would like to thank the reviewer for the valuable and constructive comments, which help us to improve the manuscript. Listed below are our point-by-point responses to the comments, including the corresponding changes made to the revised manuscript. The reviewers' comments are marked in black and our answers are marked in blue, and the revision in the manuscript is further formatted as '*Italics*'.

**Referee #1:**

This paper shows a new and innovative method of trying to represent observed changes in the 'coating' of black carbon, as detected as changes in the Dp/Dc metric produced by the DMT SP2. This works by combining an emission field with a Langrangian transport model and an empirical parameterisation for the evolution of black carbon in the atmosphere. While this 'k' parameter is tuned to the measurements, the model shows an impressive correlation with the observations, giving confidence that there is value in the technique. The authors then go on to make estimates of the contributions from inside and outside the city to the measured concentrations and properties.

While not an all-encompassing process model, this does present an intriguing new way of looking at the data that could conceivably bridge the gap between a highly detailed process model such as PARTMC-MOSAIC and Eularian chemical transport models like WRF-CHEM. It is also good that it provides another new perspective on the phenomenon of pollution 'building up' in Beijing, offering further evidence that this is driven by regional transport and transformation rather than local sources and processes. While the main innovation with this paper is technical, the authors do use it to interpret air quality sources and phenomena, so I would say this is in-scope for ACP (as opposed to AMT or GMD). While I am not completely convinced of the results (see below), I think that the novelty of the technique alone means that it deserves to be discussed within the scientific literature. Overall, the quality of English is good, but I do have certain issues regarding how the results are interpreted and presented. As such, I recommend publication after revisions.

1. General comments:

While the earlier sections of the paper were well written, I was not impressed with how the paper interpreted the implications of the results. Sections 4 and 5 in particular seem to just largely restate

what was already said in section 3 in different ways, so this entire portion of the paper could do with rewriting and sharpening up. But more generally, I would question what the key implications of this paper are; the conclusions seem to work off the reduction of Babs being a regulatory motivation, however most evidence supports BC being a more important metric for human health, which while related, is more directly related to emissions rather than processing. While the evolution of the mixing state of BC is important to consider for wet removal and radiative transfer, I would say it is debatable as to whether reducing the coatings (as opposed to overall BC) should be considered an objective for policymaking (while BC is recognized as a global climate forcing agent, the majority of this is likely from biomass burning). It is my opinion that the conclusions of this paper are more applicable to process-level atmospheric science, except for the part where the contributions to bulk BC are discussed. I would recommend the discussion and conclusions be reframed accordingly.

**Response:** Thanks for the suggestions. Following the reviewer's suggestion, we have rewritten and sharpened the Sections 4 (Discussion) and 5 (Conclusion). In terms of the Discussion section in the revised manuscript, we specifically focused on:

- presenting the simplified scheme of our model and its dependence on the model parameters,
- discussing the robustness of our model and how to extend our model calculation on BC mixing state at other seasons (e.g. summer) except for autumn/winter,
- exploring the possible mechanisms for BC aging during atmospheric transport in north china plain based on the retrieved aging rate coefficient, which is comparing with observations and those used in other models,
- implication in process-level atmospheric science.

To sharpen the conclusion of this work, we reorganized the Concluding Remarks section, as "*The effect of BC-containing particles on air quality and climate is not only dominated by BC mass concentration but also controlled by their mixing state. To better understand the mixing state of atmospheric BC in China, we developed a new approach to simulate the BC aging process during atmospheric transport based on the BC emission inventory and back-trajectory analysis. Our models track the BC mixing state (i.e., $D_p/D_c$ ratio) from an emitted source origin (e.g., a 0.25°×0.25° grid) to a receptor (i.e., Tsinghua site). The model calculation can quantify the mass-averaged $D_p/D_c$ ratio of overall BC particles over the receptor site from various*

*origins, which agreed well with observed ones. The simulations can provide information on the BC mixing state with fine temporal and spatial resolutions.*

*Based on the simulations of BC mixing state during atmospheric transport, we find a strong dependence of BC mixing state on emissions during atmospheric transport. BC particles with higher aging degrees at our site were mainly from more intensive emission origins (e.g., southern Hebei) due to higher aging rates. On the other hand, when BC particles emitted from clean origins passed through polluted regions, they were also characterized by a greater $D_p/D_c$ ratio by speeding up the aging process in polluted air. Our simulations demonstrated the importance of regional transport in BC light absorption in Beijing under polluted conditions. This provides a new perspective on the phenomenon of pollution building up in Beijing, further demonstrating that this is driven by regional transport and transformation rather than local sources and processes.."*

2. Specific comments:

(1) I found this paper extremely abstract and hard to read in places regarding certain quantities and it took me multiple reads before I think I began to understand what was going on. It particular, I would have appreciated a more intuitive explanation of what things like 'k' and 'EEI' physically represent (along with many others).

**Response:** Thanks for the comment. Following the reviewer's suggestion, we have made an explanation on a more intuitive explanation of what things like 'k' and 'EEI' physically represent. We have added the descriptions regarding the terms ('k', 'EEI', "TE" and "$(\frac{D_p}{D_c})_{ini}$") in the revised manuscript, as shown below:

*"The parameter k characterizes the rate coefficient of coating materials produced on the surface of BC by atmospheric aging such as condensation, coagulation and cloud process, which is influenced by meteorological factors, chemistry, aerosol phase state as well as other parameters (e.g., particle size)."*

*"The EEI represents the effective BC amount transported to the receptor site from the emission origins, taking into account the magnitude of BC emission from origin regions, the transport, hydrophobic-to-hydrophilic transformation, as well as dry and wet depositions during atmospheric transport."*

*"The TE is defined to quantify the transport ability of BC from origin regions to the receptor site based on transformation (i.e., hydrophobic-to-hydrophilic BC) and removal processes of BC (i.e., dry and wet depositions) in the atmospheric."*

*"The $(\frac{D_p}{D_c})_{ini}$ represents the initial the initial value of the $D_p/D_c$ ratio of BC particles before atmospheric transport, which characterizes the BC aging degree near emissions."*

(2) More discussion should be given to the possible mechanisms for the increase in Dp/Dc, i.e. coagulation and secondary aerosol formation. Given that there are already other models out there that consider these processes, is it possible to compare the value of the 'k' parameter to equivalent timescales in other models?

**Response:** Thanks. Following the reviewer's suggestion, we have added the statement on the possible mechanisms for the increase in $D_p/D_c$. In our work, the aging rate coefficient (i.e., $k_{aging}$) is determined to be ~22% h$^{-1}$, corresponding to the timescale of BC aging process about ~5 h$^{-1}$. Riemer et al. (2004) revealed that the time scale of BC aging process by condensation (2-8 h) was significantly shorter than that by coagulation (10-40 h). Higher aging rate coefficient obtained in our work indicates the BC aging in north china plain is most likely dominated by condensation process.

To the second question, the $k_{aging}$ used in our models (~22% h$^{-1}$) is comparable with the observed values in north china plain (i.e., ~20% h$^{-1}$ at Yufa site (Cheng et al., 2012) and ~21% h$^{-1}$ at Xianghe site (Zhang et al., 2018)), but significantly higher than those (1-5% h$^{-1}$) derived in developed countries and then used in models (Moteki et al., 2007; Shiraiwa et al., 2007; Cooke and Wilson, 1996; Lohmann et al., 2000; Jacobson, 2001; Koch, 2001). The difference between $k_{aging}$ used in our model and other models is consistence with the distinction between the timescale of BC aging measured in Beijing (~4.6 h corresponding to $k_{aging}$ of ~22 % h$^{-1}$) and Houston (~20 h corresponding to $k_{aging}$ of ~5 % h$^{-1}$) using an environment chamber.

Correspondingly, the related discussion has been added in the revised manuscript, as *"The retrieved aging rate coefficient $k_{aging}$ (~17% h$^{-1}$, corresponding to the timescale of BC aging process about ~6 h$^{-1}$) indicated BC particles under polluted environment at autumn/winter in NCP underwent fast aging during atmospheric transport. The retrieved $k_{aging}$ with a value of ~17% h$^{-1}$ used in our models is comparable with the observed values at other suburban sites in NCP, namely up to ~20% h$^{-1}$ and ~21% h$^{-1}$ at Yufa site and Xianghe site, respectively (Cheng et*

*al., 2012; Zhang et al., 2018). However, the aging rate coefficient commonly used in other models (1-5% h⁻¹) was significantly smaller than that used in our model (Cooke and Wilson, 1996; Jacobson, 2001; Koch, 2001; Lohmann et al., 2000;), indicating that the values used in previous models derived in developed countries (Moteki et al., 2007; Shiraiwa et al., 2007) could not represent the characteristic of BC aging process in China. A similar conclusion also found by Peng et al., (2007) using an environment chamber, namely the timescale of BC aging with significant distinction between urban cities in Beijing (~5 h corresponding to $k_{aging}$ of ~20 % h⁻¹) and Houston (~18 h corresponding to $k_{aging}$ of ~5.6 % h⁻¹). Higher aging rate coefficient suggested the BC aging under polluted environment in NCP is most likely dominated by condensation process during atmospheric transport, taking into account a clear difference between timescale of BC aging by condensation (2-8 h) and coagulation (10-40 h) (Riemer et al., 2004). "*

(3) Ultimately, while good correlation between measurement and model is reached, it is possible that other yet-to-be-identified factors may be responsible for this agreement rather than the 'k' parameter being a good representation of ageing, which is the working hypothesis. The authors should spend more time discussing how best to further test the robustness of this model. In particular, a major limitation of this work is that it is restricted to autumn/winter datasets. During the summer in Beijing, there is typically much more photochemistry (reflected in high ozone concentrations) but BC concentrations and coating thicknesses are both reduced according to Liu et al. (https://www.atmos-chem-phys-discuss.net/acp-2018-1142/). Can this result be reconciled with this model? Would it mean that the 'k' value would need to have a seasonal dependence if this model were to be extended to other months?

**Response:** Thanks to the reviewer to point this out. Following reviewer's suggestion, we have discussed how best to further test the robustness of our model. The uncertainties of our model calculation are strongly related to the retrieved parameters of aging rate coefficient ($k_{aging}$) and the initial value of $D_p/D_c$ ratio ($(\frac{D_p}{D_c})_{ini}$). It is noted that the $k_{aging}$ value (~22% h⁻¹) in this work is retrieved by the measurements under polluted environment at autumn/winter in Beijing. The retrieved $k_{aging}$ in this work can represent the characteristics of BC aging in polluted NCP at autumn/winter. In terms of BC aging at summer, the observed aging rate coefficient at other two

site (xianghe and Yufa) in NCP showed pronounced diurnal cycle with a maximum (~20% $h^{-1}$) at noon time due to stronger photochemistry (Cheng et al., 2012; Zhang et al., 2018). Figure R1 shows the aging rate coefficient of BC in the range of 0.2-21% $h^{-1}$ at summer in NCP. The maximum of $k_{aging}$ at summer noontime was comparable with that at autumn/winter. However, other time especially nighttime is significantly smaller, which can explain thinner coatings of BC at summer compared with those at autumn/winter (Liu et al., 2019).

On the other hand, the initial values of $D_p/D_c$ ratio ($(\frac{D_p}{D_c})_{ini}$) used in our model at autumn/winter are also different with that at summer. The BC transported to Beijing at autumn/winter are dominated by industrial and residential emissions with higher $D_p/D_c$ ratio near sources; however, the sources of BC in Beijing at summer are controlled by traffic and industrial emission and with smaller $(\frac{D_p}{D_c})_{ini}$, which can also explain thinner coatings of BC at summer compared with those at autumn/winter (Liu et al., 2019).

In summary, our model can be used to calculate BC mixing state during atmospheric transport at different seasons with distinct $k_{aging}$ and $(\frac{D_p}{D_c})_{ini}$) values. Both $k_{aging}$ and $(\frac{D_p}{D_c})_{ini}$) values show a seasonal dependence. Quantification of BC mixing state at different seasons using our model will be the subject of future work. In this work, we focused on the development of model.

[Figure]

Fig. S1 Diurnal cycle of BC aging rate coefficient ($k_{aging}$) at summer in Xianghe site (data from Zhang et al., (2018)).

To make this point clear, the related discussion has been added in the revised manuscript, as *"Although our model was developed and validated with autumn/winter datasets, it can be applied in other seasons with distinct model parameters (i.e. aging rate coefficient and the initial value of $D_p/D_c$ ratio). The two parameters exhibit a seasonal dependence. In terms of the aging rate coefficient at summer (Cheng et al., 2012; Zhang et al., 2018), the measurements at two suburban site (xianghe and Yufa) in NCP showed pronounced diurnal cycle in the range of 0.2-20% $h^{-1}$ with a maximum at noon time due to stronger photochemistry. The $k_{aging}$ around noontime at summer was comparable with that (~17% $h^{-1}$) at autumn/winter. However, other time especially nighttime at summer is significantly smaller. For the $(\frac{D_p}{D_c})_{ini}$, the seasonal dependence is due to different dominated sources of BC. The BC particles in Beijing at summer are controlled by traffic and industrial emission with smaller higher $D_p/D_c$ ratio near sources, but the dominated sources of BC at autumn/winter are industrial and residential emissions with higher $(\frac{D_p}{D_c})_{ini}$ (D. Liu et al., 2019; H. Liu et al., 2019). In summary, lower values both of aging rate coefficient and the initial value of $D_p/D_c$ ratio would lead to thinner coatings of BC at summer compared with those at autumn/winter in NCP, which was consistence with observations in Beijing (D. Liu et al., 2019)."*

**References**

Cheng, Y. F., Su, H., Rose, D., Gunthe, S. S., Berghof, M., Wehner, B., Achtert, P., Nowak, A., Takegawa, N., Kondo, Y., Shiraiwa, M., Gong, Y. G., Shao, M., Hu, M., Zhu, T., Zhang, Y. H., Carmichael, G. R., Wiedensohler, A., Andreae, M. O., and Pöschl, U.: Size-resolved measurement of the mixing state of soot in the megacity Beijing, China: diurnal cycle, aging and parameterization, Atmos. Chem. Phys., 12, 4477-4491, 2012.

Cooke, W. F. and Wilson, J. J. N.: A global black carbon aerosol model, J. Geophys. Res., 101, 19395–19409, 1996.

Jacobson, M. Z.: Strong radiative heating due to the mixing state of black carbon in atmospheric aerosols, Nature, 409, 695-697, 2001.

Koch, K.: Transport and direct radiative forcing of carbonaceous and sulphate aerosols in the GISS GCM, J. Geophys. Res., 106, 20311–20332, 2001.

Liu, D., Joshi, R., Wang, J., Yu, C., Allan, J. D., Coe, H., Flynn, M. J., Xie, C., Lee, J., Squires, F.,

Kotthaus, S., Grimmond, S., Ge, X., Sun, Y., and Fu, P.: Contrasting physical properties of black carbon in urban Beijing between winter and summer, Atmos. Chem. Phys., 19, 6749-6769, 10.5194/acp-19-6749-2019, 2019.

Liu, H., Pan, X., Liu, D., Liu, X., Chen, X., Tian, Y., Sun, Y., Fu, P., and Wang, Z.: Mixing characteristics of refractory black carbon aerosols determined by a tandem CPMA-SP2 system at an urban site in Beijing, Atmos. Chem. Phys. Discuss., https://doi.org/10.5194/acp-2019-244, in review, 2019.

Lohmann, U., Feichter, J., Penner, J., and Leaitch, R.: Indirect effect of sulfate and carbonaceous aerosols: a mechanistic treatment, J. Geophys. Res., 105, 12193–12206, 2000.

Moteki, N., Kondo, Y., Miyazaki, Y., Takegawa, N., Komazaki, Y., Kurata, G., Shirai, T., Blake, D. R., Miyakawa, T., and Koike, M.: Evolution of mixing state of black carbon particles: Aircraft measurements over the western Pacific in March 2004, Geophys. Res. Lett., 34, 2007.

Peng, J., Hu, M., Guo, S., Du, Z., Zheng, J., Shang, D., Levy Zamora, M., Zeng, L., Shao, M., Wu, Y.-S., Zheng, J., Wang, Y., Glen, C. R., Collins, D. R., Molina, M. J., and Zhang, R.: Markedly enhanced absorption and direct radiative forcing of black carbon under polluted urban environments, Proc. Natl. Acad. Sci. USA, 113, 4266-4271, 2016.

Riemer, N., Vogel, H., and Vogel, B.: Soot aging time scales in polluted regions during day and night, Atmos. Chem. Phys., 4, 1885-1893, 10.5194/acp-4-1885-2004, 2004.

Shiraiwa, M., Kondo, Y., Moteki, N., Takegawa, N., Miyazaki, Y., and Blake, D. R.: Evolution of mixing state of black carbon in polluted air from Tokyo, Geophys. Res. Lett., 34, L16803, doi:10.1029/2007GL029819, 2007.

Zhang, Y., Su, H., Ma, N., Li, G., Kecorius, S., Wang, Z., Hu, M., Zhu, T., He, K., Wiedensohler, A., Zhang, Q., and Cheng, Y.: Sizing of Ambient Particles From a Single-Particle Soot Photometer Measurement to Retrieve Mixing State of Black Carbon at a Regional Site of the North China Plain, J. Geophys. Res.-Atmos., 123, 12,778-712,795, doi:10.1029/2018JD028810, 2018b.

---

## Author Comment (AC2) · 24 Jun 2019

We would like to thank the reviewer for the valuable and constructive comments, which help us to improve the manuscript. Listed below are our point-by-point responses to the comments, including the corresponding changes made to the revised manuscript. The reviewers' comments are marked in black and our answers are marked in blue, and the revision in the manuscript is further formatted as '*Italics*'.

**Referee #2:**

The work developed a new approach to simulate the BC mixing state based on an emissions inventory and back-trajectory analysis. They quantified the mass-averaged aging degree of total BC particles and found the aging process of BC during atmospheric transport showed that it strongly dependent on emission levels. In general, from the modelling development to the modelling implication, I think that the result is worth to publish in the ACP after one minor revision.

1. Introduction P2L16-17 need references such as Li et al., JGR, 121(22), 13,784-713,798.

   **Response:** Thanks. We have added some related references (Li et al., 2016; Jacobson, 2001; Moffet et al., 2009; Liu et al., 2017).

2. P2L39 The author review several method to determine the BC particles. Also, Wang et al., ESTL, 2017 measure the Df in polluted air following soot particle aging. Does the model possibly consider the dimension fraction of BC? That would be more interesting.

   **Response:** Thanks for the comment. We have added some statement on change in BC morphology (quantified by dimension fraction) associated with BC mixing state under polluted conditions, as "*Moreover, Li et al., (2016) and Wang et al., (2017) revealed that the significant change in BC morphology (e.g. increase of fraction dimension) associated with their mixing state (i.e., from bare-like or partly coated to embedded BC) in polluted air, which could enhance BC light absorption (Liu et al., 2015; Cappa et al., 2012).*"

3. P7L24 deleted were P7L25 identified to indicated P7L27 import?

   **Response:** Thanks. We have revised.

4. Change the word P9L2-L3 Here it is possible to consider how RH influence BC coating? If the authors want to get the conclusion, firstly you need to exclude other possible factors. Therefore, as you try to do it.

**Response:** Thanks. In our model calculation, we just considered the effect of emissions on BC mixing state, and excluded the influences of other possible factors. A rather simplified scheme was adopted where the aging rate is assumed proportional to the emissions. Actually, the aging rate coefficient ($k$) in our models was related to meteorological factors (e.g., temperature, relative humidity (RH)), chemistry (e.g., oxidation), aerosol phase state (e.g., heterogeneous reaction) as well as other parameter (e.g., particle size). The aging rate coefficient was presumed as a constant in the model calculation, and therefore the change in BC aging degree obtained from our simulations is only caused by distinct emissions.

To make this point clear, the related statement in the revised manuscript was "*In this study, a rather simplified scheme was adopted where the aging rate is assumed proportional to the emissions without detailed consideration of the effects of temperature, particle sizes, phase state, hygroscopicity and chemistry (Riemer et al. 2009; Cheng et al. 2008, 2012, 2015; Mu et al. 2018). Actually, these factors can influence the model parameter of aging rate coefficient. Our simulations strongly depends on the emission, initial value of $D_p/D_c$ ratio and aging rate coefficient.*"

5. P9L6-13 Here some logical problem. if the aerosol in clean regions pass through the polluted area, how you know these BC don't deposit. How do you know BC at the sampling site were from clean region instead of the close polluted areas. Seemly, these are difficult questions about the mixture of air masses during the transport. How the authors separate them here?

**Response:** We thank the reviewer for raising these questions. To the first question, our model calculations consider the dry and wet deposit. In Eq. (7) in the manuscript, the *TE* represents the transport efficiency, taking into account transformation, transport, and removal processes of BC in the atmosphere. The TE is be treated as a weighting factor to quantify effective amount of BC from the emission origin to receptor site.

To the second question, we can separate BC from various emission origins (i.e., 0.25°×0.25° grid). Here, we discussed the gridded $D_p/D_c$ ratio, representing the aging degree of BC particles that are transported to the receptor site from a certain origin (i.e., a grid $h$) rather than all origins.

Our model calculation can track the evolution of BC aging degree from a certain origin (i.e., a grid h) to the receptor site throughout different regions.

6. P10-P11, the implications and discussion should be combined? The implications should not be in Result section. The authors need to consider the structure in the two section.

Response: Thanks. We have changed "3.3 Implication for BC light absorption" into "BC light absorption in the atmosphere"

7. P12L10-11, capability should be individual BC or total here? My question is that the statement could be one problem. If you think total capability of BC in upper layer is higher that of BC in the lower. You need to know the BC distribution in the column within PBL. Normally the BC concentration is much higher than them in upper layer. Even mixing state is higher in upper layer but their concentration is lower. Therefore, the total capacity need be questioned.

Response: Thanks. Sorry for the misleading and here the capability of BC light absorption represents the light absorption per unit BC mass, namely mass cross section (MAC) of BC. The MAC of BC strongly depends on their mixing state due to lens effect of coating materials on BC surface. The MAC and BC mass concentration ($C_{BC}$) are two factors to determine BC light absorption ($\sigma_{ab}$), namely $\sigma_{ab}=MAC \times C_{BC}$.

To make it clear, we have changed the "*capability of BC light absorption*" into "*capability of BC light absorption (i.e., mass cross section of BC-containing particles including coating materials on BC surface)*".

8. P12 section 5 concluding remarks. Most of them repeat the results or others. Is that possible to shorten it? The details should not be concluded here such as P13L2-5 and so on. The reference should be removed in the section.

Response: Thanks. Following the reviewer's suggestion, we have sharpened the Concluding Remarks section, as "*The effect of BC-containing particles on air quality and climate is not only dominated by BC mass concentration but also controlled by their mixing state. To better understand the mixing state of atmospheric BC in China, we developed a new approach to simulate the BC aging process during atmospheric transport based on the BC emission*

[revised manuscript text omitted]